# Let Brain Rhythm Shape Machine Intelligence for Connecting Dots on Graphs

**Jiaqi Ding**[1]*    **Tingting Dan**[2]*    **Zhixuan Zhou**[1]    **Guorong Wu**[1,2]†
[1]Department of Computer Science    [2]Department of Psychiatry
The University of North Carolina at Chapel Hill
{jiaqid, zzhixuan}@cs.unc.edu   {Tingting_Dan, grwu}@med.unc.edu

*"The whole universe is a complex system, including the human brain."* – Immanuel Kant

## Abstract

In both neuroscience and artificial intelligence (AI), it is well-established that neural "coupling" gives rise to dynamically distributed systems. These systems exhibit self-organized spatiotemporal patterns of synchronized neural oscillations, enabling the representation of abstract concepts. By capitalizing on the unprecedented amount of human neuroimaging data, we propose that advancing the theoretical understanding of rhythmic coordination in neural circuits can offer powerful design principles for the next generation of machine learning models with improved efficiency and robustness. To this end, we introduce a physics-informed deep learning framework for B̲rain R̲hythm I̲dentification by K̲uramoto and C̲ontrol (coined *BRICK*) to characterize the synchronization of neural oscillations that shapes the dynamics of evolving cognitive states. Recognizing that brain networks are structurally connected yet behaviorally dynamic, we further conceptualize rhythmic neural activity as an artificial dynamical system of coupled oscillators, offering a shared mechanistic bridge to brain-inspired machine intelligence. By treating each node as an oscillator interacting with its neighbors, this approach moves beyond the conventional paradigm of graph heat diffusion and establishes a new regime of representation compression through oscillatory synchronization. Empirical evaluations demonstrate that this synchronization-driven mechanism not only mitigates over-smoothing in deep GNNs but also enhances the model's capacity for reasoning and solving complex graph-based problems.

## 1 Introduction

The evolution of artificial intelligence (AI) has long been intertwined with efforts to decipher the principles of human cognition. Rather than existing as separate disciplines, AI and neuroscience share conceptual foundations in how information is represented and transformed. In the human brain, massive ensembles of neurons form densely interconnected circuits that coordinate activity across multiple spatial and temporal scales [4]. Many artificial graph systems, ranging from transportation networks to online communities, show comparable organizational motifs, including efficient long-range connectivity and modular substructures [57].

Both domains can be viewed through the lens of collective dynamics. Neural assemblies give rise to oscillatory rhythms through intricate coupling between regions [20], while graph neural

---

*Equal contribution.
†Corresponding author.

39th Conference on Neural Information Processing Systems (NeurIPS 2025).

networks (GNNs) update node states by iteratively exchanging information, leading to structured representations that emerge across layers [58].

Machine learning shows remarkable proficiency in extracting statistical patterns from vast datasets, yet it remains fundamentally limited in its capacity for flexible reasoning and context-aware integration. Biological neural systems, in contrast, constantly fuse inputs from different sensory streams in real time [24]. Everyday experiences, for example, hearing approaching footsteps before seeing someone, highlight how sensory predictions interact across modalities to construct unified percepts.

**Neural oscillations** serve as a core mechanism that enables coordinated information exchange across different sensory modalities [53, 49]. At the microscopic scale, accumulating evidence indicates that neurons communicate through lateral connections [31], and their rhythmic patterns of excitability, commonly referred to as brain rhythms, emerge from the intrinsic dynamics of local circuits and the biophysical properties of ionic channels [8]. Neighboring neurons often synchronize their activity, forming competitive clusters that interpret sensory inputs [25, 45]. This process, known as "competitive learning" [2], compresses information during layer-wise propagation and enhance abstraction. Such synchronization also drives functional specialization among higher-order cortical populations, aligning their activity (analogous to fireflies synchronizing their flashes) to generate compact and meaningful neural representations [7].

While modern GNNs such as graph convolution network (GCN) [34] and Graph Transformer [63] excel at structural representation learning, they lack the adaptive dynamical properties that characterize biological neural systems. On the flip side, cognition in humans and animals is governed by large-scale, dynamic oscillations that support perception, memory and decision-making. This parallel between biological and artificial systems motivates us to rethink graph learning through the lens of coupled neural oscillators, which presents new opportunities to reshape GNN architectures.

Following this spirit, we aim to establish a novel *learning mechanism for graph-structured data* grounded in neural oscillatory synchronization, as illustrated in Fig. 1. ***First***, we introduce *BRICK*, a physics-informed deep learning framework for brain rhythm identification. This model unifies the synchronization dynamics of the Kuramoto model [36] with the concept of attending memory [32], allowing us to cap-

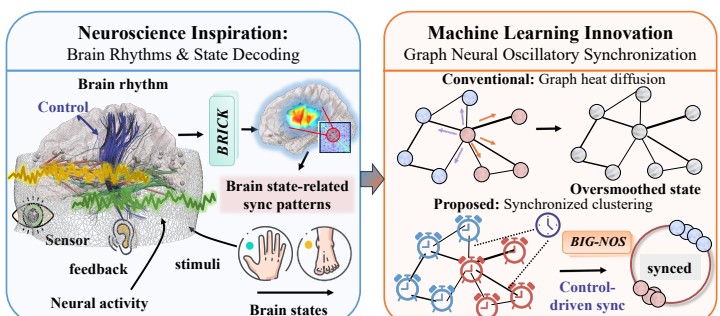

Figure 1: The link between the human brain and artificial intelligence inspires the development of novel learning mechanism of graph-structured data based on neural oscillatory synchronization.

ture rhythmic neural oscillations that underlie cognitive state transitions. By bridging control theory with oscillatory dynamics, *BRICK* provides a principled foundation for extracting interpretable task-related synchronization patterns from complex neural signals. ***Second***, we generalize this oscillatory synchronization principle to graph learning and propose *BIG-NOS*, which is a biologically inspired graph neural oscillatory synchronization framework. Here, each graph node functions as a coupled oscillator whose state evolves dynamically through network interactions. In contrast to conventional graph message passing and convolution paradigms constrained by heat diffusion [11], which often suffer from over-smoothing and information dilution [37, 17], *BIG-NOS* preserves oscillatory coherence across graph topology. By steering feature propagation via controlled synchronization, the model naturally captures emergent interference patterns in the spatial domain, unveiling meaningful modular structures (e.g., red–blue clustering in Fig. 1). This biologically grounded formulation draws inspiration from how rhythmic synchronization supports information integration in the human brain, enriching the theoretical connection between brain dynamics and graph learning and offering a powerful foundation for next-generation machine intelligence, capable of interpretable reasoning, robust pattern integration, and scalable performance on real-world graph datasets. In summary, we propose a neural oscillatory synchronization-based graph learning mechanism, uniting a physics-informed model (*BRICK*) and a biologically grounded GNN (*BIG-NOS*) with decent real-world performance.

## 2 Related Works

**Graph neural networks for brain network modeling.** GNNs have become foundational tools for analyzing structured data, spanning areas such as chemistry, recommendation systems, biological networks, and social graphs. Initial developments focused on spectral formulations like GCN [34] and GAT [56], which were later extended by more expressive variants including GIN [59] and GraphSAGE [27]. Beyond standard tasks of node and graph classification, GNNs have been adapted for spatio-temporal modeling [60], link prediction, and graph generation [38], with scalability improved through methods such as clustering and sparse attention [15, 61, 35].

Applications to brain networks have followed a similar trajectory, using both structural and functional connectomes to represent the complex topology of neural systems. Early work such as Brain-NetCNN [33] demonstrated the feasibility of disease classification from brain graphs, while more recent studies leverage hierarchical pooling [62] and attention mechanisms [29] to uncover multi-scale functional organization and enhance interpretability. Temporal extensions [60] have further enabled dynamic modeling of fMRI and EEG signals, offering insight into evolving neural states over time.

**System identification approaches.** A key objective in computational neuroscience has been to uncover mechanistic principles governing how the brain regulates perception, action, and cognition [6]. Motivated by parallels between cognitive control and classical control theory, prior work has represented neural dynamics as linear or bilinear control systems [26, 40], enabling controllability analyses through energy-based formulations [41]. More recently, research has shifted toward integrating recurrent neural architectures with physics-inspired continuous-time dynamics [14, 28, 16]. The Kuramoto model, in particular, has been a central tool for linking structural connectivity to emergent functional activity [9, 10], showing that spontaneous neural fluctuations can be explained as coordinated interactions between local oscillators and large-scale network architecture. Our proposed *BRICK* builds on this line of work by parameterizing these governing dynamics through a deep learning framework.

**Kuramoto dynamics in machine learning.** A closely related effort is the recent development of artificial Kuramoto oscillatory neurons for unsupervised vision representation [43], which leverage synchronization to achieve compact and structured feature embeddings. In contrast, our approach emphasizes the bidirectional exchange between neuroscience and machine learning through dynamical systems principles [18], and places additional focus on probing the intrinsic potential of the model. Specifically, we aim to (1) use machine learning models to reveal the mechanisms by which fluctuating brain activity maps to cognitive states, and (2) translate biologically grounded synchronization mechanisms into graph learning to enable large-scale network reasoning without explicit supervision.

## 3 Methods

In this work, we model both human brain networks and graph-structured data as dynamical systems whose collective behavior is governed by the Kuramoto model [18]. Consider a network of $N$ interacting oscillators with pairwise coupling strengths $K_{ij}$ $(i, j = 1, \ldots, N)$. Each oscillator oscillates at its intrinsic frequency $\omega_i$, and the temporal evolution of its phase $\theta_i$ can be expressed as

$$\frac{d\theta_i}{dt} = \omega_i + \sum_{j=1}^{N} K_{ij} \sin\left(\theta_j - \theta_i\right), \tag{1}$$

where $\theta_i \in \mathbb{R}$ denotes the instantaneous phase of the $i$-th oscillator. When coupling is absent, the oscillators evolve independently following their own natural frequencies. As interactions accumulate over time, the sinusoidal coupling term in Eq. 1 drives gradual phase alignment across units, giving rise to collective synchronization. This interaction enables groups of oscillators to entrain to common rhythms, ultimately forming stable phase-locked clusters (as shown in Fig. 1 bottom-right).

### 3.1 Deep Model for Brain Rhythm Identification

A growing body of research has highlighted the central role of neural oscillations in coordinating activity across distributed brain regions [8, 55]. To mechanistically characterize these large-scale synchronization phenomena, the Kuramoto model has become a widely used framework in both theoretical neuroscience and neuroimaging research [9].

**Problem setup.** We formalize the brain network as a weighted graph $\mathcal{G} = (V, W)$, where $V$ represents $N$ brain regions (nodes) and $W = [w_{ij}]_{i,j=1}^{N}$ denotes the weighted adjacency matrix. Each weight $w_{ij}$ encodes the coupling intensity between node $v_i$ and $v_j$, derived from neuroimaging measurements[3]. Let the observed BOLD signal be $X = x_i \mid x_i(t) \in \mathbb{R}, t = 1, \ldots, T$, where $x_i(t)$ is the signal measured from brain region $v_i$ at time $t$. We then construct a physics-informed deep learning model to map neural activity patterns to cognitive outcomes.

**Vectorized neural oscillators.** To move beyond scalar-valued signals $x_i(t)$ and better characterize complex neural dynamics, we adopt the geometric scattering transform (GST) [23]. GST leverages harmonic wavelets derived from the graph Laplacian to construct representations that encode both multi-scale structure and frequency content. We begin by defining the lazy random walk matrix [42]: $G = \frac{1}{2}(I_N + WD^{-1})$, where $I_N$ denotes the $N \times N$ identity matrix and $D$ is the diagonal degree matrix associated with $W$. From this operator, a sequence of graph wavelets $\{\Psi_h\}_{h=0}^{H-1}$ is generated recursively, enabling a compact and structured embedding of neural oscillatory activity across multiple scales. $\Psi_0 := I_N - G, \quad \Psi_h := G^{2^{h-1}} - G^{2^h}, \quad 1 \le h < H$. Let $x(t) \in \mathbb{R}^N$ be the BOLD snapshot at time $t$. For each scale $h$, the GST output $\hat{x}^h(t) \in \mathbb{R}^N$ is computed by: (1) applying the wavelet transform $(\Psi_h, x(t))$, (2) taking element-wise absolute value, and (3) applying the low-pass filter $\Phi = G^{2^H}$, yielding $\hat{x}^h(t) = \Phi|(\Psi_h, x(t))|$.

**Vectorized Kuramoto model for multi-frequency neural synchronization.** Following GST decomposition, each brain region $v_i$ yields a set of frequency-resolved BOLD signals $\hat{x}_i(t) = [\hat{x}_i^h(t)]_{h=1}^{H}$ capturing temporal fluctuations across multiple scales. The evolution of these oscillator phases can be compactly described by a vector-valued differential equation:

$$\frac{d\hat{x}_i}{dt} = \omega_i + [\tau \cdot \phi_{\hat{x}_i}(\sum_{j=1}^{N} w_{ij}\hat{x}_j)] \tag{2}$$

where $\hat{x}_i \in \mathbb{R}^H$ denotes the oscillator state of region $v_i$ on a unit hypersphere, $\omega_i$ its intrinsic frequency, and $\tau$ a global coupling gain parameter. The mapping $\phi$ constrains the aggregated phase input to the tangent space of $\hat{x}_i$, enforcing smooth manifold-constrained dynamics.

Rather than simply fitting temporal fluctuations, the aim is to represent neural synchronization in a way that reflects how the brain achieves coordinated oscillatory activity with minimal energetic cost. Accordingly, we seek an oscillator-based representation and a control strategy that are both mathematically stable and biologically meaningful.

**Kuramoto model with attending memory.** The standard Kuramoto model (Eq. 2) relies on global dynamics, which are not well suited for capturing short-lived functional fluctuations tied to specific cognitive operations. In contrast, the human brain can flexibly emphasize task-relevant inputs while down-weighting irrelevant signals, effectively deciding "what to remember" and "what to ignore." This selective filtering resonates with the concept of attending memory [32] and motivates us to augment the Kuramoto model with a biologically inspired, memory-guided control mechanism. We incorporate a global optimal control term:

$$\frac{d\hat{x}_i}{dt} = \omega_i + [\tau \cdot \phi_{\hat{x}_i}(y_i + \sum_{j=1}^{N} w_{ij}\hat{x}_j)], \tag{3}$$

where $y_i$ is a feedback control signal derived from attending memory, reflecting observed neural activity. Each $y_i$ represents behavior-specific, population-level memory traces that dynamically modulate the influence of oscillator $i$ in a given cognitive context. This control formulation allows the system to continuously adapt synchronization dynamics as neural states evolve.

**Intuition behind the Kuramoto model with attending memory.** Assuming uniform contribution from each brain region $v_i$ and a symmetric coupling matrix $W$, the extended Kuramoto dynamics in Eq. 3 can be viewed as a gradient flow that evolves along the steepest descent direction of the following energy functional $E$ [3, 43]:

$$E = -\sum_{i,j} \hat{x}_i^{\top} w_{ij} \hat{x}_j - \sum_{i} y_i^{\top} \hat{x}_i \tag{4}$$

The term $E$ acts as a Lyapunov function that guarantees the system's stability, as $\frac{dE(\hat{x}(t))}{dt} \le 0$. Its first component captures pairwise coupling, promoting synchronization among oscillators $\hat{x}_i$ and $\hat{x}_j$

---

[3]For example, $w_{ij}$ can represent the normalized fiber count in structural connectivity (SC) or the strength of temporal co-fluctuation in functional connectivity (FC).

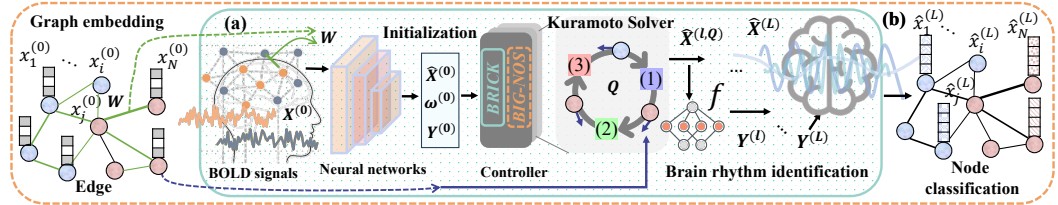

Figure 2: Overview of the proposed framework. (a) *BRICK* (solid cyan box) models neural oscillatory synchronization for brain rhythm identification. (b) *BIG-NOS* (dashed orange box) extends the same Kuramoto-based dynamics to graph learning.

proportional to the interaction strength $w_{ij}$. The second component represents task-driven feedback control, aligning each oscillator's dynamics with its control signal $y_i$, which encodes population-level attention and behavior-specific modulation. Together, these two objectives describe a biologically motivated trade-off between network coherence and cognitive specialization.

From a control-theoretic standpoint, the feedback variables $Y = \{y_i | i = 1, ..., N\}$ can be interpreted as spatial control fields, providing a physically grounded attention mechanism distinct from graph attention [56]. These feedback patterns quantify the controllability of each brain region [39], bridging the link between oscillatory synchronization and adaptive regulation of cognitive states. Importantly, the energy formulation also offers a gradient-flow interpretation, ensuring that the system monotonically decreases $E$ along its trajectory, thereby maintaining smooth and stable evolution on the hyperspherical manifold.

This unified view reveals why the proposed model alleviates the over-smoothing problem observed in deep GNNs: oscillatory synchronization fosters global coherence while preserving discriminative, high-frequency information. Consequently, this mechanism provides both a neurobiologically meaningful and computationally efficient pathway toward large-scale graph reasoning, forming the theoretical basis of the *BIG-NOS* model introduced in Sec. 3.2.

**Physics-informed deep model for brain rhythm identification.** Building upon the above formulation, we introduce *BRICK*, a physics-informed framework designed to predict cognitive states through neural synchronization dynamics. The overall architecture of the model is illustrated in the solid cyan box in Fig. 2a. Specifically, the intrinsic frequency of each oscillator is determined by a set of learnable vectors $\Gamma_\sigma(\hat{x}_i)$, whose magnitudes control the rotation speed, while the control signal $y_i$ is generated through a neural mapping $\Gamma_\mu(\hat{x}_i)$, with $\sigma$ and $\mu$ denoting network parameters. The synchronization process evolves in three key steps: **(1) Coupling influence aggregation.** For each oscillator $\hat{x}_i$, we compute the aggregate input from its neighbors as $z_i = y_i + \sum_{j=1}^{N}(s_{ij} \cdot w_{ij})\hat{x}_j$, where $S = [s_{ij}]_{i,j=1}^{N}$ is a symmetric, trainable reweighting matrix that adaptively modulates the coupling strengths $W$. This step captures how local interactions shape the global phase landscape. **(2) Tangent-space projection.** Each $z_i$ is then projected onto the tangent space of the unit hypersphere at $\hat{x}_i$ through $\phi_{\hat{x}_i}(z_i) = z_i - \langle z_i, \hat{x}_i \rangle \hat{x}_i$. This ensures that the updated direction respects the geometric constraint of the manifold and maintains stability during the synchronization process [12]. **(3) Phase-state update.** Finally, $\hat{x}_i$ is updated via forward Euler integration of Eq. 3, while the controller $y_i$ is transformed using a mapping function $f_\varphi$ to remain on the hypersphere and capture phase-invariant patterns. These three steps together implement a geometry-aware synchronization mechanism in which oscillator states evolve under both coupling dynamics and task-driven control. The procedure is summarized in Algorithm 1.

The downstream objective combines the task-driven loss $\mathcal{L}$, defined as cross-entropy between predicted and ground-truth cognitive task labels, with the synchronization energy $E$. For unseen subjects, the trained *BRICK* synchronizes a large oscillator ensemble $\hat{X}$ using learned parameters $\sigma$ and $\mu$, producing both the predicted cognitive state and its associated control pattern $Y$.

### 3.2 *BIG-NOS*: Graph Learning via Neural Oscillation

**New learning mechanism of GNN.** Conceptually, *BIG-NOS* extends *BRICK* from continuous brain dynamics to arbitrary graph data (as shown in the dashed orange box of Fig. 2b). By treating graph nodes as analogs of brain regions and the adjacency matrix as a proxy for inter-regional coupling

**Algorithm 1:** Iterative Solver for *BRICK* Dynamics (Eq. 3)

---

**Input:** BOLD signal $X^{(0)}$, coupling matrix $W$
**Output:** Final oscillator states $\hat{X}^{(L)}$ and controller $Y^{(L)}$
**Initialization:** Obtain initial oscillator states $\hat{X}^{(0)}$ by applying GST to $X^{(0)}$; set
  $Y^{(0)} = \Gamma_\mu(X^{(0)})$;
Parameterize natural frequencies $\Omega \leftarrow \Gamma_\sigma(\hat{X}^{(0)})$;
**for** $l = 1$ **to** $L$ **do**
  **for** $q = 1$ **to** $Q$ **do**
    // Step 1: Aggregate coupling influence
    $Z \leftarrow Y^{(l)} + (S \odot W)\hat{X}^{(q)}$;
    // Step 2: Project to tangent space of the hypersphere
    $\phi_{\hat{X}^{(q)}}(Z) \leftarrow Z - \langle Z, \hat{X}^{(q)} \rangle \hat{X}^{(q)}$;
    // Step 3: Integrate oscillator dynamics
    $\Delta \hat{X}^{(q)} \leftarrow \Omega + \tau \cdot \phi_{\hat{X}^{(q)}}(Z)$;
    $\hat{X}^{(q+1)} \leftarrow \hat{X}^{(q)} + \beta \cdot \Delta \hat{X}^{(q)}$;
    // Step 4: Renormalize to stay on manifold
    $\hat{X}^{(q+1)} \leftarrow \hat{X}^{(q+1)}/\|\hat{X}^{(q+1)}\|$;
  **end**
  // Step 5: Update control signal on hypersphere
  $Y^{(l+1)} \leftarrow f_\varphi(\hat{X}^{(q+1)})$;
**end**

---

strength, we seamlessly adapt the physics-informed architecture of *BRICK* to general graphs. Both models share the same governing equation rooted in Kuramoto synchronization but differ in their data domains and learning objectives.

Let $X = [x_i]_{i=1}^N$ denote the initial graph embeddings. In conventional GNNs, feature propagation follows a graph heat diffusion process, formalized as $\frac{\partial X}{\partial t} = \nabla \cdot (\nabla X)$, where $\nabla$ and $\nabla\cdot$ denote the graph gradient and divergence operators [11]. Prior studies have shown that excessive message passing under this diffusion mechanism leads to over-smoothing, where node representations become indistinguishable [37]. Inspired by *BRICK*, we mitigate over-smoothing by evolving graph features within a latent oscillatory phase space rather than diffusing them over the graph domain. The Kuramoto model's intrinsic oscillatory dynamics, driven by coupling interactions and intrinsic frequencies, naturally prevent convergence to static equilibrium (as in diffusion-based GNNs [51]) and instead produce partial synchronization [1].

Unlike conventional attention mechanisms that re-weight neighbor contributions during aggregation [56], our model introduces *control patterns $Y$* as task-dependent feedback signals that guide synchronization dynamics (Eq. 3). These patterns provide a new perspective on graph controllability through the lens of complex systems theory [39].

Taken together, conventional GNNs are typically governed by a diffusion equation, in which node features evolve according to a heat-diffusion process. This formulation leads to passive information averaging across neighboring nodes, an approach that effectively captures local smoothness but often suffers from over-smoothing and limited expressiveness in deeper architectures. In contrast, our brain-rhythm-inspired *BIG-NOS* is formulated as a neural oscillation system, where each node behaves as a coupled oscillator whose state evolves according to both its intrinsic frequency and the phase interactions with connected nodes. Rather than diffusive averaging, information propagation in *BIG-NOS* emerges from oscillatory synchronization, allowing it to represent complex, nonlinear dependencies across the network. Furthermore, unlike conventional attention mechanisms that rely on fixed weighting matrices, *BIG-NOS* incorporates a task-adaptive feedback control mechanism that dynamically modulates coupling strength, enabling flexible coordination and improved generalization across diverse graph structures.

**Network architecture.** We discretize the *BRICK* formulation into a GNN framework, denoted as *BIG-NOS*, to enable general graph learning. In this architecture, each node is represented as an

oscillator, and its interactions are governed by the underlying graph topology. The model takes initial graph embeddings $X^{(0)}$ as input, and an encoder (e.g., a GCN layer) is used to initialize the control term. As illustrated in Fig. 2b, feature evolution in *BIG-NOS* follows Kuramoto-based dynamics, which are explicitly described in Algorithm 1.

# 4    Experiments

In this section, we present extensive experiments to validate the effectiveness and interpretability of our proposed models across a variety of brain-related and general graph learning tasks.

We conduct a comprehensive evaluation of our proposed models across multiple tasks. We evaluate *BRICK* for brain rhythm identification and explore its effectiveness in unsupervised brain parcellation, demonstrating its potential for uncovering latent neural patterns in neural dynamics. We further evaluate *BIG-NOS* on standard benchmarks including node and graph classification. To analyze its robustness to "over-smoothing", we examine performance across increasing network depths. We compared *BIG-NOS* with a diverse set of graph-based baselines: GCN [34], GAT [56], GIN [59], GCNII [13], GraphSAGE [27], SAN [35], GRAND [11], GTN [63], GraphCON [48] and KuramotoGNN (KGNN) [46]. To validate scalability, we also evaluate *BIG-NOS* on the large-scale ogbn-arxiv dataset. For this task, we compare not only against the above baselines but also with top-performing models from the official OGB leaderboard.

## 4.1    Data Description

To evaluate the effectiveness of *BRICK* and *BIG-NOS*, we use the publicly available neuroimaging and graph datasets, respectively.

● **Human brain datasets**

*1. HCP-Aging (HCP-A) [5].* This dataset includes 717 subjects with 4,846 fMRI scans (300 time points each), covering four tasks (VISMOTOR, CARIT, FACENAME, Resting State).

*2. HCP-Young Adults (HCP-YA) [54].* This includes seven cognitive tasks: Motor, Relational, Social, Working Memory, Language, Emotion, and Gambling (175 time points each). The Working Memory task (HCP-WM) involves alternating 2-back and 0-back conditions with body, place, face, and tool stimuli with a total of 405 time points. Preprocessing follows [19].

In both HCP-A and HCP-YA datasets, the brain is parcellated into 116 regions via the AAL atlas [52]. Structural connectivity (SC) matrices with the size of $116 \times 116$ encode fiber counts between regions (normalized per subject), while functional connectivity (FC) is derived from Pearson correlations of BOLD signals across brain regions. To test scalability, we use the Brainnetome atlas [22] to increase granularity to 246 regions for HCP-WM. We perform 5-fold cross-validation for both the 4-class (HCP-A), 7-class (HCP-YA) and 8-class (HCP-WM) classification tasks.

● **Graph-based datasets**

*1. Node classification.* We apply *BIG-NOS* to homophilic and heterophilic graphs sorted by homophily ratio $h$ [64]: Texas ($h$=0.11), Wisconsin (0.21), Actor (0.22), Squirrel (0.22), Chameleon (0.23), Cornell (0.3), Citeseer (0.74), Pubmed (0.80), Cora (0.81). We also evaluate on the large-scale **ogbn-arxiv** dataset from the Open Graph Benchmark (OGB) [30]. Details are in Table 4 (Appendix).

*2. Graph classification.* We evaluate on ENZYMES and PROTEINS from TUDataset [44], detailed in Table. 5. For homophilic graph data (Cora, Citeseer, Pubmed) in node classification tasks, we adopt the semi-supervised 20-per-class training split from [34] and average over 5 random seeds. For heterophilic graph data and TUDataset, we follow [47] and [21], using 10-fold cross-validation and reporting test-set averages. All datasets are evaluated using Accuracy (Acc), weighted-Precision (Pre) and F1-Score (F1).

## 4.2    Performance on Human Brain Data

### 4.2.1    Brain rhythm identification

*Results.* We evaluate *BRICK* on the task of decoding dynamic brain states through neural synchronization. Fig. 3a reports the performance across three human brain datasets and seven baseline

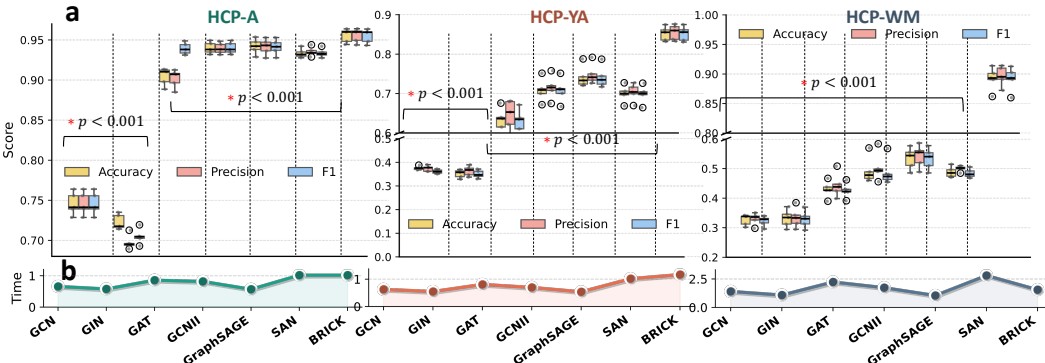

Figure 3: (a) Performance (%) on human brain data. $^*$ indicates statistically significant improvement ($p <$ 0.001). (b) Inference time (ms/subject) on three human brain datasets.

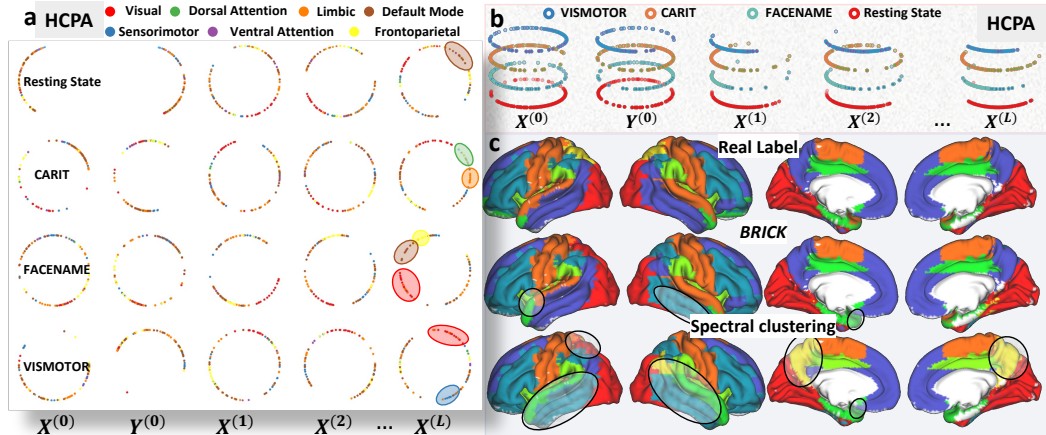

Figure 4: (a) Distribution of regional features across different brain states (tasks) in the HCP-A dataset. Each brain region's feature trajectory over time is projected onto the unit phase space (magnitude = 1), revealing consistent synchronization of the same regions across tasks. (b) Phase-space feature aggregation on HCP-A dataset, where features from different categories at each time point are mapped onto the unit phase space to illustrate their clustering patterns. (c) Unsupervised clustering comparison between *BRICK* and classical spectral clustering, where *BRICK* yields more coherent and neurobiologically meaningful functional communities.

models. *BRICK* consistently achieves the best performance ($p < 0.001$), surpassing all existing hand-designed GNN models.

***Discussion.*** These results demonstrate that our physics-based oscillation model effectively synchronizes brain regions, producing patterns closely aligned with specific cognitive states. To further explore this relationship, we visualize the phase-space representations of brain regions in Fig. 4a. *BRICK* captures task-specific synchronization across functionally defined brain network in HCP-A. For instance, Visual network regions (red) cluster clearly during the VISMOTOR task in the deeper representation $X^{(L)}$, indicating strong within-network coordination. Similarly, tighter phase clustering emerges in the Sensorimotor and Dorsal Attention networks under tasks such as CART and FACENAME. These results suggest that our model respects functional boundaries in a biologically meaningful manner.

We also investigate task-specific synchronization across all subjects. As shown in Fig. 4b, individuals performing the same task (e.g., "VISMOTOR" or "FACENAME" in HCP-A dataset) form well-separated synchronization in deeper layers (e.g., $X^{(L)}$). This indicates *BRICK* captures latent dynamics consistent across subjects and datasets, supporting the notion that neural phase alignment underlines cognitive state representation.

### 4.2.2 Potential in unsupervised brain parcellation

To evaluate our model's capability in unsupervised settings, we examine functional parcellation on the HCP-A dataset, which partitions 116 brain regions (AAL atlas) into eight functional subnetworks

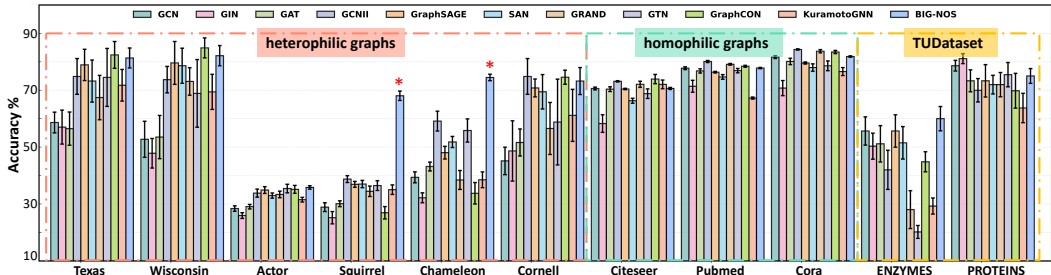

Figure 5: Accuracy (%) on node and graph classification tasks. $^{*}$ indicates significant improvement ($p < 0.001$).

(default mode, frontoparietal, limbic, ventral/dorsal attention, sensorimotor, visual, and cerebellar networks). We train an *unsupervised* variant of *BRICK* that identifies functional communities by minimizing pairwise distances between synchronized oscillators $x_i$ and $x_j$ using a Rayleigh quotient objective [50]. As shown in Fig. 4c, compared to classical spectral clustering, *BRICK* more effectively synchronizes related brain regions into meaningful clusters, demonstrating strong potential for personalized brain parcellation.

### 4.2.3 Ablation study and model efficiency

We conduct an ablation study on the GST to assess the impact of different frequency scales. Specifically, we vary the wavelet order (from 0 to 2) and level (1 or 2) on the HCP-YA dataset (fixed-group split), which define a multi-branch wavelet tree: the level determines the tree's depth, while the order sets the branching factor. As shown in Table 1, increasing either the level or order generally improves performance by incorporating additional frequency components. However, this comes at the cost of increased computational complexity. For example, using `level=2` and `order=[0,1]` expands the feature dimension from $[N]$ to $[N, 7]$, due to the increased number of wavelet filters.

Table 1: Ablation results of GST with different levels and orders on HCP-YA.

| Level | Order | Acc (%) | Pre (%) | F1 (%) |
|---|---|---|---|---|
| 1 | [0] | -0.41 | -0.45 | -0.42 |
| | [0, 1] | -0.14 | -0.05 | -0.16 |
| | [0, 1, 2] | 85.20 | 85.53 | 85.13 |
| 2 | [0] | -0.23 | -0.24 | -0.24 |
| | [0, 1] | +0.21 | +0.25 | +0.22 |

We provide a detailed comparison of inference time per subject across three human brain datasets in Fig. 3b. Overall, all baseline GNNs demonstrate low computational cost, with inference typically below 2 ms per subject. We further PDE-based GRAND and KuramotoGNN are notably slower. In contrast, *BRICK* achieves comparable efficiency to lightweight baselines such as GCN and GraphSAGE. Even on the larger HCP-WM dataset (246 regions), *BRICK* maintains a modest runtime of 1.56 ms per subject (as shown in Fig. 3b-the rightmost data point), confirming its scalability and computational practicality for large-scale brain analysis.

### 4.3 Performance on Graph Data

### 4.3.1 Node classification

***Result.*** Fig. 5 (left and middle) presents the results of eleven models across nine node-classification benchmarks. Overall, *BIG-NOS* achieves the best or second-best accuracy on the majority of datasets, showing strong generalization across both homophilic and heterophilic graphs. For homophilic graphs (*Citeseer*, *Pubmed*, *Cora*), *BIG-NOS* maintains competitive performance on par with advanced architectures like GCNII and GRAND. For heterophilic graphs, it achieves the highest performance on *Actor*, *Squirrel*, and *Chameleon*, where conventional message-passing GNNs typically fail due to inconsistent neighborhood features. In contrast, *BIG-NOS* continues to improve through phase-based coupling, which allows nodes to synchronize dynamically rather than relying solely on feature similarity. This oscillatory mechanism enables coherent message propagation even when node attributes are diverse, making *BIG-NOS* especially suitable for heterophilic settings.

*Scalability.* To assess the scalability of *BIG-NOS*, we evaluated its performance on the large-scale ogbn-arxiv dataset. As reported in Table 2, our model achieved an accuracy of **0.86**, outperforming

Table 2: Performance on **ogbn-arxiv** test set. **Bold** and underline indicate the best and second-best results.

| Dataset | Metric | GCN | GIN | GAT | GCNII | GraphSAGE | SAN | BiGTex[a] | SimTeG+TAPE+RevGAT[a] | *BIG-NOS* |
|---------|--------|-----|-----|-----|-------|-----------|-----|-----------|------------------------|-----------|
| **ogbn-arxiv** | Acc | 0.70 | 0.71 | 0.70 | 0.71 | 0.71 | 0.69 | **0.89** | 0.78 | 0.86 |
| | Pre | 0.70 | 0.70 | 0.69 | 0.71 | 0.70 | 0.68 | – | – | **0.86** |
| | F1 | 0.69 | 0.69 | 0.68 | 0.70 | 0.69 | 0.68 | – | – | **0.85** |

[a] *BiGTex* and *SimTeG+TAPE+RevGAT* are the top two models on the OGB Leaderboard at the time of writing. Pre and F1 are not reported.

all standard baselines and ranking second on the leaderboard, just behind BiGTex (0.89). These results underscore the strong generalization ability of our oscillatory synchronization mechanism on large graphs.

***Discussion.*** These results show that our message aggregation strategy effectively promotes synchronization among nodes of the same class while preserving the structural dependencies defined by the coupling strength. In contrast to conventional GNNs, which tend to drive neighboring nodes toward indistinguishable embeddings, our model maintains representational diversity and prevents excessive feature homogenization. As a result, it inherently resists over-smoothing as the network depth increases. This property is empirically verified in Fig. 6, where we observe consistently high classification performance even with up to 128 layers, demonstrating the robustness of our synchronization framework against the depth-induced degradation common in traditional GNNs.

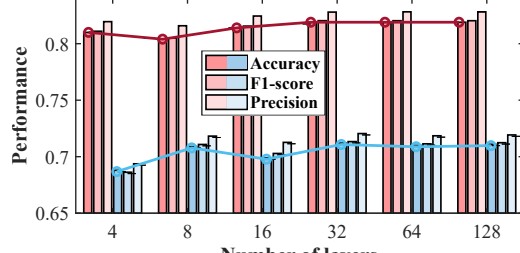

Figure 6: Performance of *BIG-NOS* as the network depth increases on *Cora* (red) and *Citeseer* (blue).

### 4.3.2 Graph classification

***Result.*** Fig. 5 (right) reports graph classification accuracy on the ENZYMES and PROTEINS datasets from TUDataset [44]. *BIG-NOS* achieves the best accuracy on ENZYMES ($60.00_{\pm 4.28}$) and performs competitively on PROTEINS ($75.02_{\pm 2.61}$), despite using a minimal batch size (batch size=1). These results indicate that our model maintains strong generalization from node-level to graph-level tasks without architectural modification.

***Discussion.*** Graph classification is conceptually analogous to brain rhythm identification, where the goal is to predict a global state emerging from local interactions. The success of *BIG-NOS* in this setting further validates the ability of our synchronization mechanism to generate global patterns that effectively distinguish different graphs, which reinforces its applicability beyond node-level tasks.

Table 3 summarizes the average accuracy across homophilic, heterophilic, and all datasets. The balanced performance is particularly encouraging: whereas many existing models tend to specialize in a single regime (for example, GCN performs well primarily on homophilic graphs), our model demonstrates consistent generalization across both homophilic and heterophilic graphs. This robustness suggests that the underlying neural oscillation mechanism is inherently generalizable and largely independent of specific graph structures.

Table 3: The average accuracy across homophilic, heterophilic, and all datasets.

| | GCN | GIN | GAT | GCNII | GraphSAGE | SAN | GRAND | GTN | GraphCON | KGNN | *BIG-NOS* |
|---|-----|-----|-----|-------|-----------|-----|-------|-----|----------|------|-----------|
| **Hete. Avg** | 42.19 | 39.46 | 43.99 | 62.01 | 58.19 | 57.18 | 50.55 | 54.99 | 56.28 | 51.23 | **69.18** |
| **Homo. Avg** | 76.68 | 66.81 | 75.77 | 79.20 | 75.46 | 72.99 | 78.32 | 74.77 | 78.61 | 71.96 | **76.76** |
| **Total Avg** | 53.69 | 48.58 | 54.58 | 67.74 | 63.95 | 62.45 | 59.80 | 61.58 | 63.73 | 58.14 | **71.70** |

## 5 Conclusion

In this work, we introduce *BRICK*, a new deep learning framework inspired by the biological mechanisms underlying fluctuating brain activity and cognitive state formation. Building on this foundation, we extended the concepts of neural oscillatory synchronization and attentional memory to graph domain and derive a novel graph-neural architecture, *BIG-NOS*. Extensive experiments demonstrate that *BIG-NOS* effectively alleviates over-smoothing problem in conventional GNNs, and delivers competitive performance across diverse GNN benchmarks. These results highlight the model's promise for tackling challenging graph-learning problems while offering a biologically grounded view of information propagation.

## Acknowledgement

This work was supported by the National Institutes of Health (AG091653, AG068399, AG084375) and the Foundation of Hope. The views and conclusions contained in this document are those of the authors and should not be interpreted as representing the official policies, either expressed or implied, of the NIH.

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

# A Technical Appendices and Supplementary Material

## A.1 Data Description

Table 4: Dataset description for node classification.

|              | Texas | Wisconsin | Actor  | Squirrel | Chameleon | Cornell | Citeseer | Pubmed | Cora  | ogbn-arxiv |
|--------------|-------|-----------|--------|----------|-----------|---------|----------|--------|-------|------------|
| Hom. ratio $h$ | 0.11  | 0.21      | 0.22   | 0.22     | 0.23      | 0.3     | 0.57     | 0.74   | 0.81  | -          |
| # Nodes      | 183   | 251       | 7,600  | 5,201    | 2,277     | 183     | 3,327    | 19,717 | 2,708 | 169,343    |
| # Edges      | 295   | 466       | 26,752 | 198,493  | 31,421    | 280     | 4,676    | 44,327 | 5,278 | 1,166,243  |
| # Classes    | 5     | 5         | 5      | 5        | 5         | 5       | 7        | 3      | 6     | 40         |

Table 5: Dataset description for graph classification.

|             | ENZYMES | PROTEINS |
|-------------|---------|----------|
| Avg # Nodes | 32.63   | 39.36    |
| Avg # Edges | 62.14   | 39.06    |
| # Classes   | 6       | 2        |
| # Graphs    | 600     | 1,113    |

## A.2 Hyperparameters

As human data as an example, for all methods, the hidden dimension is set to 256. The network depth and batch size are set to 2 and 64 for the comparison methods, while for our *BRICK*, they are set to 16 and 256, respectively.

## A.3 Connections and Distinctions from Prior Approaches

### A.3.1 Relevance to neuroscience domains

In principle, the learning behavior of *BRICK* is shaped by the same governing equations (i.e., Kuramoto model) that describe the neural oscillatory dynamics responsible for generating cognition and behavior. We further introduce structural-functional coupling, implemented through the geometric scattering transform, to emulate the cross-frequency coupling observed in cognitive neuroscience. In addition, we incorporate an adaptive control term to enhance task-relevant modulation. Drawing inspiration from neural oscillatory synchronization, our *BRICK* model not only improves predictive accuracy but also enhances interpretability, thereby offering novel insights into cognitive processes.

### A.3.2 Advances over conventional methods in brain rhythm identification

While traditional models, including CNNs, RNNs, Transformers, and GNNs, have demonstrated strong empirical performance in brain rhythm identification, they largely remain data-driven and agnostic to the underlying neurobiological mechanisms, they often treat the brain as a generic data source without sufficient domain knowledge. In contrast, *BRICK* explicitly incorporates neural synchronization dynamics, a well-established principle in cognitive neuroscience, into the learning architecture. This alignment with neuroscience allows us to interpret learned patterns (e.g., coupling strengths or synchronization clusters) in biologically meaningful terms (as shown in Figure 3), such as inter-regional communication or pathological desynchronization. Therefore, compared to conventional approaches, *BRICK* is not just a deep model for feature representation learning, it shows the potential to establish a biologically inspired reasoning system. By embedding core principles of brain dynamics into a learnable, differentiable architecture, our proposed deep model enables achieving more interpretable, temporally-aware, and robust brain state identification.

### A.3.3 Differences from prior dynamical models

While both *BRICK* and Cabral *et al.* [9] adopt the Kuramoto model to explore brain dynamics, their roles and applications diverge significantly. In their work, the Kuramoto model is used as a forward simulator to reproduce biologically observed functional connectivity (FC) patterns from empirical structural connectivity (SC), with the goal of explaining emergent brain phenomena. In contrast, our *BRICK* repurposes the Kuramoto model as a computational module within a machine learning framework. Rather than passively simulating dynamics, *BRICK* actively learns to perform representation learning, reasoning, and predictive tasks on graphs using dynamic synchronization mechanisms.

For methodological paradigm, while Capouskova *et al.* [10] follow a data-driven analytical paradigm, using autoencoders and clustering to uncover latent cognitive states from empirical fMRI data, they encode BOLD phase coherence data using modern machine learning tools without Kuramoto model. Their focus lies in neuroscientific pattern discovery and interpretation. In contrast, *BRICK* introduces a physics-informed learning framework that embeds the principles of neural synchronization, specifically via the Kuramoto model, directly into the learning process. The outcomes of these two approaches are also distinct: Capouskova *et al.*[10] generate empirical insights about cognitive brain states, whereas *BRICK* yields new machine learning models that not only excel in brain-related tasks but also generalize to broader graph-based AI problems.

Compared with [43], our *BRICK* framework offers two key advantages. First, neuroscientific grounding: starting from fMRI-derived brain rhythms, we augment the Kuramoto core with a task-driven feedback controller that mimics cognitive control, thereby preserving biological interpretability. Second, domain-specific impact: on real HCP fMRI data *BRICK* achieves the best task decoding and unsupervised parcellation, while on graph benchmarks it delivers competitive accuracy and remains resistant to oversmoothing even at 128 layers.

Taken together, these properties make *BRICK* a biologically inspired but practically efficient module that bridges brain dynamics and general-purpose AI.

## A.4 Discussion and Limitation

Although our approach demonstrates promising accuracy on brain connectomes and node-level benchmarks, its computational efficiency remains limited due to the iterative solver required to integrate *BRICK* dynamics. Unlike feed-forward GNNs, our model relies on repeated geometric projections, oscillator updates, and manifold renormalization, all of which introduce substantial per-batch computational overhead. This bottleneck becomes particularly pronounced on datasets such as TUDataset.

Despite these efficiency constraints, our method still achieves competitive graph classification performance, underscoring the robustness of our dynamical-systems perspective beyond neuroscience applications. In future work, we plan to develop more efficient numerical solvers and adaptive graph batching strategies to significantly improve scalability. We also aim to apply our framework to broader multimodal and clinical datasets to further assess its generalizability and translational value.

More broadly, this line of research highlights the potential of brain-inspired synchronization mechanisms for building more interpretable and principled AI systems.

More broadly, this line of research highlights the promise of brain-inspired synchronization mechanisms for building more interpretable and efficient AI systems, while also drawing attention to important ethical considerations surrounding the use of neural data and the societal impact of autonomous decision-making technologies.

