# OpenReview forum: "Let Brain Rhythm Shape Machine Intelligence for Connecting Dots on Graphs"
_NeurIPS.cc/2025/Conference — NeurIPS 2025 poster_

### Official Review · Reviewer_Rarc · 2025-06-24

**Clarity:** 3
**Significance:** 3
**Originality:** 3
**Rating:** 5
**Confidence:** 3

**Summary:**

This paper proposes a new learning framework called BRICK. It develops more effective machine learning algorithm design principles by simulating brain rhythm mechanisms, thereby connecting neuroscience and artificial intelligence. The goal of BRICK is to identify brain rhythms, and to develop an oscillator representation and control mechanism closely integrated with fundamental neuroscience principles. The representations output by this model highly reflect neuroscientific principles (i.e., brain rhythms). Furthermore, building upon this, the paper extends this concept to the graph domain, introducing a new graph neural network architecture called BIG-NOS. This model improves existing graph learning models and possesses the ability to avoid the over-smoothing issue.

**Questions:**

See weaknesses. Overall, I believe this paper is a good work. I’m happy to raise my score if the authors address my concerns well.

**Ethical Concerns:**

["NO or VERY MINOR ethics concerns only"]

**Final Justification:**

The authors have addressed my concerns, so I'd like to raise my score to 5.

**Limitations:**

See weaknesses.

**Quality:**

3

**Strengths And Weaknesses:**

Strengths:
1. This paper presents novel and significant work. It introduces a new perspective to the AI field, particularly in graph neural networks, by proposing a biologically inspired mechanism of brain rhythms for machine learning.
2. This paper experimentally demonstrates the model's ability to identify brain rhythms. For example, BRICK can decode dynamic brain states based on neural synchronisation patterns and achieves significantly superior performance compared to existing GNN models on human brain datasets such as HCP-A, HCP-YA, and HCP-WM.
3. This paper effectively avoids the over-smoothing issue in GNNs. The proposed BIG-NOS explicitly avoids over-smoothing by evolving graph features within a latent oscillatory phase space, rather than through traditional heat diffusion methods.

Weaknesses:
1. The paper does not compare with popular models like graph diffusion and graph transformer, thus lacking a comprehensive comparison with the latest state-of-the-art graph learning models [1-3].
2. The interpretability (biological plausibility) analysis is insufficient. In brain rhythm identification, the authors demonstrate visualisation results of brain region synchronisation, but do not provide further analysis on whether the output representations are consistent with biological characteristics of the brain. For instance, Line 305 states that the proposed mode uncovers functional communities that are not
only align with known subnetworks but also capture novel patterns in brain organisation, but a more detailed biological interpretation is lacking.
3. The paper lacks theoretical proof for its core mechanism (e.g., how oscillatory synchronisation mitigates over-smoothing). Although an energy function $E$ is proposed, it does not provide complete evidence to support its theoretical results. Although $E$ is claimed to be a "natural" Lyapunov function, no detailed mathematical derivations or proofs are given.

[1] Yang, H., Wang, B., & Jia, J. (2024). Gnncert: Deterministic certification of graph neural networks against adversarial perturbations. In The Twelfth International Conference on Learning Representations.

[2] Chamberlain, B., Rowbottom, J., Gorinova, M. I., Bronstein, M., Webb, S., & Rossi, E. (2021, July). Grand: Graph neural diffusion. In International conference on machine learning (pp. 1407-1418). PMLR.

[3] Yun, S., Jeong, M., Kim, R., Kang, J., & Kim, H. J. (2019). Graph transformer networks. Advances in neural information processing systems, 32.

---

> ### Author Rebuttal · Authors · 2025-07-31
>
> We sincerely thank the reviewer for recognizing the novelty and importance of our work. We also greatly appreciate the constructive feedback regarding **empirical performance**, **neuroscientific interpretability**, and **theoretical grounding** of the proposed model. Below, we provide detailed responses to each of the reviewer’s concerns.
>
> ### *1. Lacking a comprehensive comparison with the latest state-of-the-art graph learning models [1-3].*
>
> **[Comparison with additional graph learning methods]**
> We have added two baseline models as reviewer suggested: GRAND, GTN. In addition, we also included KuramotoGNN (KGNN)[1] and GraphCON [2], which are both recent graph models for oscillatory dynamics. While we initially considered GNNCert, it focuses on adversarial robustness and its reported clean accuracy is based on existing backbones GCN, GAT, and GIN, which are already part of our benchmark. Nonetheless, we would include GNNCert to the related work to acknowledge its relevance in the broader graph learning. The updated results are presented in Table below, the results show that our models still achieve significant improvements in brain datasets and retain strong performance in general graph benchmarks.
>
> **Brain states identification:**
> |Acc(%)|HCP-A|HCP-YA|HCP-WM|
> |-|--|-|-|
> |GRAND|86.18±1.79|47.36±0.67|33.27±2.63|
> |GTN|82.56±0.87|53.42±1.21|30.68±0.36|
> |GraphCON|87.87±2.29|66.29±1.04|44.12±2.1|
> |KGNN|85.53±1.68|45.68±1.12|35.79±1.54|
> |BRICK|95.55±0.77|84.20±1.60|89.22±1.71|
>
> **Node classification:**
> |Acc(%)|Texas|Wisconsin|Actor|Squirrel|Chameleon|Cornell|Citeseer|Pubmed|Cora|
> |-|-|-|-|-|-|-|-|-|-|
> |GRAND|67.42±7.81|73.09±4.83|33.34±1.16|34.45±1.84|38.43±3.33|56.55±9.15|72.1±1.07|79.12±0.3|83.73±0.58|
> |GTN|74.52±10.2|68.89±11.9|35.42±1.5|36.45±1.72|55.84±4.08|58.81±15.09|68.8±1.69|76.90 ± 0.77|78.60±1.72|
> |GraphCON|82.43±4.72|83.72±4.48|35.13±1.38|26.9±2.17|33.75±3.77|74.59±2.48|62.9±5.95|71.21±7.36|62.73±5.79|
> |KGNN|71.75±5.57|69.43±6.18|31.5±0.82|35.02±1.67|38.5±2.78|61.16±9.16|72.06±1.53|67.24±0.36|76.57±1.4|
> |BIG-NOS|81.35±3.51|82.16±3.56|35.8±0.6|68.06±1.65|74.5±1.13|73.24±4.75|70.63±0.36|77.8±0.23|81.86±0.25|
>
> **Graph classification:**
> |Acc(%)|ENZYMES|PROTEINS|
> |-|-|-|
> |GRAND|28±6.67| 71.97±4.26|
> |GTN|20.15±2.27| 75.48±4.27|
> |GraphCON|44.83±3.53| 69.82±6.12|
> |KGNN|29.26±2.84| 63.76±5.17|
> |BIG-NOS|60±4.28| 75.02±2.61|
>
> ### *2. The interpretability (biological plausibility) analysis is insufficient.*
>
> **[More detailed interpretability (biological plausibility) analysis]**
> We now elaborate how the discovered patterns in Figure 3 support neurobiological plausibility:
> * **Alignment with functional subnetworks (Figure 3a):**
> *BRICK* captures task-specific synchronization across functionally defined brain network in HCP-A. For instance, Visual network regions (red) cluster clearly during the VISMOTOR task in the deeper representation $X^{(L)}$, indicating strong within-network coordination. These results suggest that our model respects functional boundaries in a biologically meaningful manner.
> * **Intersubject consistency and task-relevant encoding (Figure 3b):**
> This figure shows that individuals performing the same task (e.g., “SOCIAL” in HCP-YA) form well-separated synchronization in deeper layers $X^{(L)}$. This indicates BRICK captures latent dynamics consistent across subjects, supporting the notion that neural phase alignment underlines cognitive state representation.
> * **Unsupervised Functional Differentiation (Figure 3c):**
> Here, we assess whether BRICK can produce meaningful cortical parcellations without supervision. Compared to spectral clustering, BRICK yields phase-based partitions that are more spatially localized and better aligned with known functional networks [3]. This supports the claim that BRICK captures biologically coherent dynamics.
>
> In this way, BRICK offers a mechanistic perspective grounded in neural oscillation theory, which allows us to move beyond black-box representations toward interpretable latent dynamics.
>
> ### *3. The paper lacks theoretical proof for its core mechanism (e.g., how oscillatory synchronisation mitigates over-smoothing). Although E is claimed to be a "natural" Lyapunov function, no detailed mathematical derivations or proofs are given.*
>
> **[Why *E* is a *natural* Lyapunov function?]**
> 1. **Physical intuition**
>
> The dynamical system (Eq.(4))
> $
> \frac{d\hat{x}\_i}{dt} = \omega_i + \gamma \phi\_i( y_i + \sum\_{j=1}^N w\_{ij} \hat{x}\_j)
> $
> is a network of weakly-coupled oscillators driven by an external signal $y_i$. This suggests the following energy function (Eq.(5)):
>
> $
> E(\hat{x}) = -\sum\_{i,j} \hat{x}\_i^\top w\_{ij} \hat{x}\_j - \sum\_i y_i^\top \hat{x}\_i
> $,
>
> where the first sum represents the **coupling term** and the second sum is the **driving term**. Because the update $\frac{d\hat{x}_i}{dt}$ explicitly attempts to align $\hat{x}_i$ with its neighbors $(w\_{ij} \hat{x}\_j)$ and with $y_i$, the quantity measuring *how well aligned* they are, namely \$E\$, is the physically most natural Lyapunov candidate.
>
>  2. **Lyapunov property** [4]
>
> Let $\hat{\mathbf{x}} = [\hat{x}\_1, ..., \hat{x}\_N] \in \mathbb{R}^{N \times d}$ and recall dynamical system Eq.(4) and energy $E$ Eq.(5). Because $W = [w\_{ij}]$ is symmetric (undirected),
>
> $
> \nabla_{\hat{\mathbf{x}}} E = - (W \hat{\mathbf{x}} + \mathbf{y}), \quad \mathbf{y} = [y_1 \ldots y_N]
> $
>
> Define $u_i := y_i + \sum_j w\_{ij} \hat{x}\_j$. Then Eq.(4) may be rewritten as:
>
> $
> \dot{\hat{x}}_i = \omega_i + \gamma \phi\_i(u_i) = \omega\_i - \gamma \phi\_i(-u\_i) = \omega\_i - \gamma \phi\_i([\nabla\_{\hat{\mathbf{x}}} E]\_i)
> $
>
> Then the time derivative of $E$ is defined as:
>
> $
> \dot{E} = \left\langle \nabla\_{\hat{\mathbf{x}}} E, \dot{\hat{\mathbf{x}}} \right\rangle = \sum\_i \underbrace{ \langle [\nabla\_{\hat{\mathbf{x}}} E]\_i, \omega_i \rangle}\_{\text{(i)}} - \gamma \underbrace{\langle [\nabla\_{\hat{\mathbf{x}}} E]\_i, \phi\_i([\nabla\_{\hat{\mathbf{x}}} E]\_i) \rangle }\_{\text{(ii)}}
> $
>
> The $\omega_i$ induces a pure phase rotation; it is orthogonal to the gradient direction, so
> $
>  \langle[\nabla_{\hat{\mathbf{x}}}E]_i, \omega_i\rangle=0
> $
>
> And $\phi\_i(\cdot)$ is
>   **(i)** odd: \$\phi\_i(-z) = -\phi\_i(z)\$, and
>   **(ii)** monotone increasing:
> $
>  \langle \phi_i(z_1) - \phi_i(z_2), z_1 - z_2 \rangle \geq 0
> $.
> With \$z = \[\nabla\_{\hat{\mathbf{x}}} E]\_i\$, we have \$\langle z, \phi\_i(z) \rangle \geq 0\$, this therefore yields (ii) $\leq 0\$. Combining (i) and (ii),
>
> $
> \dot{E}=0-\gamma\sum_i\langle z, \phi_i(z)\rangle\leq0, \quad z=[\nabla_{\hat{\mathbf{x}}} E]_i
> $
>
> If \$W \succeq 0\$ and both \$w\_{ij}\$ and \$y\_i\$ are finite, then Eq.(5) is at most quadratic with a finite lower bound:
>
> $
> E(\hat{\mathbf{x}})\geq-\lambda_{\max}(W) \|\hat{\mathbf{x}}\|^2-\|\mathbf{y}\| \|\hat{\mathbf{x}}\| \geq c_1 \|\hat{\mathbf{x}}\|^2-c_2
> $,
> for suitable constants $c_1 > 0$, $c_2 \geq 0$.
>
> Eq.(5) and the bound above provide the classical Lyapunov conditions:
> $
> E(\hat{\mathbf{x}}) \geq c_1 \|\hat{\mathbf{x}}\|^2 - c_2, \quad \dot{E}(\hat{\mathbf{x}}) \leq 0
> $
>
> Hence **\$E\$ is a Lyapunov function** for the Eq.(4): it is lower-bounded and monotonically non-increasing along every trajectory, guaranteeing stability.
>
> **[Heuristic Analysis: Why *BRICK* alleviates over-smoothing]**
>
> We analyze why the *BRICK* can theoretically alleviate the oversmoothing by examining its simplified dynamics, steady-state solution and spectral response.
>
> **Simplified BRICK Dynamics.**
> To derive an interpretable steady-state solution, we consider a simplified version of BRICK dynamics. We assume a constant natural frequency $\omega_i$ across all nodes and linearize the nonlinear projection $\phi$:
>
> $
> \frac{d\hat{\mathbf{x}}}{dt} = -\hat{\mathbf{x}}+W\hat{\mathbf{x}}+\mathbf{y}
> $
>
> This corresponds to a linear consensus-like system, where the $-\hat{\mathbf{x}}$ term mimics a dissipative force, ensuring stability.
>
> **Equilibrium Solution.**
> At steady state $( \frac{d\hat{\mathbf{x}}}{dt} = 0)$, we obtain:
>
> $
> (I - W)\hat{\mathbf{x}}^* = \mathbf{y} \quad \Rightarrow \quad \hat{\mathbf{x}}^* = (I - W)^{-1} \mathbf{y}
> $
>
> The inverse exists provided the spectral radius of $W$ is strictly smaller than 1 (i.e., $\lambda_k \neq 1, \forall k$), a condition that is typically satisfied for normalized adjacency or attention matrices used in practice.
>
> **Spectral Interpretation.**
> Let $W$ be symmetric and decomposed as $W = U \Lambda U^\top$, where $U$ is the orthonormal eigenvector matrix and
> $\Lambda = \text{diag}(\lambda_1, ..., \lambda_n)$. Projecting into the spectral domain:
>
> $
> \hat{\mathbf{x}}^* = U(I - \Lambda)^{-1}U^\top \mathbf{y}
> $
>
> Let $\tilde{\mathbf{y}} = U^\top \mathbf{y}$ and $\tilde{\hat{\mathbf{x}}}^* = U^\top \hat{\mathbf{x}}^*$, then:
>
> $
> \tilde{\hat{x}}^*_k = \frac{1}{1 - \lambda_k} \tilde{y}_k
> $
>
> Thus, each spectral mode is scaled by a transfer function:
> $
> h(\lambda_k) = \frac{1}{1 - \lambda_k}
> $
>
> **Comparison with Diffusion-based GNNs.**
> In standard diffusion-type GNNs (e.g., GCN), applying $L$ layers is equivalent to using a transfer function
> $e^{-L\lambda_k}$ in spectral space. This leads to exponential suppression of high-frequency signals (large $\lambda_k$), causing oversmoothing. In contrast, *BRICK* uses a transfer function that decays much slower:
> $h(\lambda_k) = \frac{1}{1 - \lambda_k}$, which corresponds to $1/\lambda$-level suppression, allowing high-frequency, discriminative signals to persist.
>
> [1] Rusch, T. K. et al. (2022). Graph-coupled oscillator networks.
> [2] Nguyen, T. et al. (2024). From coupled oscillators to graph neural networks: Reducing over-smoothing via a kuramoto model-based approach.
> [3] Yeo, B. T. T. et al. (2011). The organization of the human cerebral cortex estimated by intrinsic functional connectivity.
> [4] Takeru M. et al. Artificial kuramoto oscillatory neurons.
>
> **Thanks again for your feedback, if the reviewer has additional questions, we would be glad to provide further clarifications.**

---

> > ### Comment · Reviewer_Rarc · 2025-08-04
> >
> > Thank you for your response. I have updated my score.

---

> > > ### Author Response · Authors · 2025-08-05
> > >
> > > Thank you very much for updating the score. We truly appreciate your time and thoughtful feedback throughout the review process.

---

### Official Review · Reviewer_zxED · 2025-07-01

**Clarity:** 3
**Significance:** 3
**Originality:** 3
**Rating:** 5
**Confidence:** 3

**Summary:**

This framework presents a physics and neuroscience-informed deep learning framework. It integrates with a synchronization mechanism of neural oscillations inside a graph representation learning framework. Motivated by addressing the oversmoothing issue of conventional GNNs, it proposed to utilize the brain rhythms in the artificial dynamical system. This method integrates a Kuramoto model with attending memory for modeling oscillatory synchronization in brain regions. It has been applied to solve two tasks on brain rhythm identification and conventional tasks on graph data, demonstrating superior performance compared to existing baselines.

**Questions:**

1. What might be challenges for the optimization solver for the proposed graph learning procedure, would this guarantee to converge, and how it is sensitive to initialization and hyperparameters?
2. Would this be more computational expensive compared to conventional graph learning methods, and how it will be scale to larger graph?

**Ethical Concerns:**

["NO or VERY MINOR ethics concerns only"]

**Final Justification:**

Thanks for the authors' detailed responses. I am especially impressed by the scientific insights and motivations clarified by the authors. I also appreciates the mechanistic understanding on neuronal synchronization that this work could bring. The responses on baselines comparisons, initialization, convergence also mostly addressed my concerns. I would like to increase my score to accept (5). Thanks for your efforts.

**Limitations:**

1. Include more discussions with neuroscience literatures and mechanisms would be more impactful.
2. Include more discussion and evaluations on the computational cost, and scalability.

**Paper Formatting Concerns:**

Not applied.

**Quality:**

3

**Strengths And Weaknesses:**

**Strengths**
1. This work presents a novel solution to address the oversmoothing issue of conventional GNNs, it incorporated with a physics-informed structure with Kuramoto model with attending memory inside a graph representation framework. This system has novel designs in governing equation, learning mechanism, and attention mechanism compared to conventional GNN models.
2. The model demonstrates superior performance on two tasks (brain rhythm identification and graph tasks) compared to existing baselines.
3. This model with good interpretability, and demonstrate interesting visualizations of synchronization patterns in Fig 3.
4. Multiple ablations studies are conducted.

**Weaknesses**
1. It would be more impactful to link the synchronization mechanisms to neuroscience domains, and explains why it might be more superior compared to conventional approachs in brain rhythm identification task, and how the discovery in Fig 3 might be relevant to existing studies.
2. Missing comparisons of the computational cost of the proposed methods compared to existing methods.

---

> ### Author Rebuttal · Authors · 2025-07-30
>
> We sincerely thank the reviewer for recognizing the novelty and comprehensive nature of our work. We also greatly appreciate the reviewer’s interest and thoughtful feedback regarding the **biological interpretability**, **computational efficiency** and **scalability**, as well as the **optimization dynamics** underlying our framework. Below, we address each of the reviewer’s concerns in detail.
>
> ### _1. It would be more impactful to link the synchronization mechanisms to neuroscience domains, and explains why it might be more superior compared to conventional approachs in brain rhythm identification task, and how the discovery in Fig 3 might be relevant to existing studies._
>
> **[Link to neuroscience domains]**
> In principle, the learning behavior of _BRICK_ is shaped by the same governing equations (i.e., Kuramoto model) that describe the neural oscillatory dynamics responsible for generating cognition and behavior. We further introduce structural-functional coupling, implemented through the geometric scattering transform (Line 159), to emulate the cross-frequency coupling observed in cognitive neuroscience. In addition, we incorporate an adaptive control term to enhance task-relevant modulation. Drawing inspiration from neural oscillatory synchronization, our BRICK model not only improves predictive accuracy but also enhances interpretability, thereby offering novel insights into cognitive processes (Section 3.1).
>
> **[Advances compared to conventional approaches in brain rhythm identification task]**
> While traditional models, including CNNs, RNNs, Transformers, and GNNs, have demonstrated strong empirical performance in brain rhythm identification, they largely remain data-driven and agnostic to the underlying neurobiological mechanisms, they often treat the brain as a generic data source without sufficient domain knowledge. In contrast, _BRICK_ explicitly incorporates neural synchronization dynamics, a well-established principle in cognitive neuroscience, into the learning architecture. This alignment with neuroscience allows us to interpret learned patterns (e.g., coupling strengths or synchronization clusters) in biologically meaningful terms (as shown in Figure 3), such as inter-regional communication or pathological desynchronization. Therefore, compared to conventional approaches, _BRICK_ is not just a deep model for feature representation learning, it shows the potential to establish a biologically inspired reasoning system. By embedding core principles of brain dynamics into a learnable, differentiable architecture, our proposed deep model enables achieving more interpretable, temporally-aware, and robust brain state identification.
>
> **[Relevance of Figure 3 to existing studies]**
> Below is a detailed explanation of how the discovered patterns in Figure 3 align with known neurobiological structures and functions:
> * **Alignment with functional subnetworks (Figure 3a):**
> *BRICK* captures task-specific synchronization across functionally defined brain network in HCP-A. For instance, Visual network regions (red) cluster clearly during the VISMOTOR task in the deeper representation $X^{(L)}$, indicating strong within-network coordination. Similarly, tighter phase clustering emerges in the Sensorimotor and Dorsal Attention networks under tasks such as CART and FACENAME. These results suggest that our model respects functional boundaries in a biologically meaningful manner.
> * **Intersubject consistency and task-relevant encoding (Figure 3b):**
> This figure shows that individuals performing the same task (e.g., “SOCIAL” or “EMOTION” in HCP-YA) form well-separated synchronization in deeper layers (e.g., $X^{(L)}$). This indicates BRICK captures latent dynamics consistent across subjects and datasets, supporting the notion that neural phase alignment underlines cognitive state representation.
> * **Unsupervised Functional Differentiation (Figure 3c):**
> Here, we assess whether BRICK can produce meaningful cortical parcellations without supervision. Compared to traditional spectral clustering, BRICK yields phase-based partitions that are more spatially localized and better aligned with known functional networks [1]. This supports the claim that BRICK captures biologically coherent dynamics.
>
> By mapping learned feature vectors onto a phase manifold, we can interpret inter-regional phase alignment as a proxy for functional coordination. In this way, _BRICK_ offers a mechanistic perspective grounded in neural oscillation theory, which allows us to move beyond black-box representations toward interpretable latent dynamics.
>
> ### _2. Missing comparisons of the computational cost of the proposed methods compared to existing methods. Would this be more computational expensive compared to conventional graph learning methods, and how it will be scale to larger graph?_
>
> **[Comparisons of the computational cost]**
> We have provided a detailed comparison of the inference time per subject for all models across three brain datasets in **Appendix A.2 Table 7**. In addition, we have now included a summary table that reports the average inference time of each model on all brain datasets. As shown in the results, our proposed model **BRICK** has slightly higher inference time compared to lightweight baselines such as **GCN**, **GIN**, and **GraphSAGE**. However, it remains more efficient than several other models, while also delivering significantly better performance on brain datasets.
>
> | Model                           | GCN | GIN  | GAT  | GCNII | GraphSAGE | SAN  | BRICK |
> | ------------------------------- | --- | ---- | ---- | ----- | --------- | ---- | ----- |
> | Avg inference time (ms/subject) | 0.9 | 0.74 | 1.31 | 1.09  | 0.72      | 1.62 | 1.25  |
>
> **[How to scale to larger graph]**
> To ensure the scalability of our method to large-scale graphs, we implement all matrix operations using sparse representations. This design allows our model to scale comparably to lightweight models such as GCN, and enables full-batch training on large graphs like ogbn-arxiv without exceeding memory limits.
>
> Moreover, for even larger graphs, e.g., fine-grained applications such as vertex-level cortical modeling in neuroscience, our framework can readily incorporate subgraph sampling strategies [2] to enable smooth training.
>
> ### _3. What might be challenges for the optimization solver for the proposed graph learning procedure, would this guarantee to converge, and how it is sensitive to initialization and hyperparameters?_
>
> **[Initialization and hyperparameters ablations]**
> **Sensitivity to initialization.**
> To assess the sensitivity of our model to random initialization, we conducted experiments on the Cora, Citeseer, and Pubmed datasets using **5** different random seeds. As reported in Table 2, our model achieves highly consistent results, with the maximum standard deviation across these runs being only **0.36%**. This indicates that our method is stable and not significantly affected by initialization variance.
>
> **Sensitivity to hyperparameters.** We conducted **several** ablation studies on the most critical hyperparameters and reported them in the appendix.
>
> * In **Appendix A.2**, we ablate two parameters specific to the **BRICK** architecture: *L* (outer iterations for control update) and *Q* (inner time steps for oscillator dynamics). Increasing both parameters increases inference time, while having only a marginal effect on accuracy. This suggests the model is not overly sensitive to these parameters within a reasonable range.
>
> * In **Appendix A.3**, we further analyze the sensitivity of geometric scattering transform (GST)-related hyperparameters: *level* and *order*. Our results show that increasing either improves performance by capturing more frequency components, but at the cost of slightly increased computation.
>
> **[Optimization challenge and converge guarantee]**
> We appreciate the reviewer’s insightful question regarding the optimization behavior of our proposed method. From a theoretical perspective, the dynamics of our model are governed by a *gradient flow* is:
> $
> \frac{d\hat{x}_i}{dt}=\omega_i+\gamma\cdot \phi_i (y_i+\sum\_{j=1}^{N} w\_{ij}\hat{x}_j)
> $,
>
> which corresponds to a gradient descent flow on the following energy landscape:
> $
> E = - \sum_{i,j} w_{ij} \hat{x}_i^\top \hat{x}_j - \sum_i y_i^\top \hat{x}_i
> $.
> This energy function $E$ acts as a natural Lyapunov function, satisfying the condition: $\frac{dE}{dt} \leq 0$,
> thus ensuring that the system always evolves towards a (local) steady state. Therefore, the entire system remains a valid gradient flow on a constrained manifold, which theoretically guarantees convergence.
>
> In practice, we also observe stable optimization behavior during training. To support this claim, we will include the **training loss trajectories** of our model on representative datasets in **Appendix**.
>
> [1] Yeo, B. T. T. et al (2011). The organization of the human cerebral cortex estimated by intrinsic functional connectivity. Journal of Neurophysiology, 106(3), 1125–1165.
> [2] Zeng, H. et al (2019). Graphsaint: Graph sampling based inductive learning method. arXiv preprint arXiv:1907.04931.
>
> Thanks again for your feedback, if anything is still unclear or the reviewer would like additional clarification, we would gladly continue the discussion and address any further concerns.

---

### Official Review · Reviewer_SVVA · 2025-07-01

**Clarity:** 2
**Significance:** 4
**Originality:** 4
**Rating:** 5
**Confidence:** 3

**Summary:**

The paper presents BRICK, a novel deep learning-based framework designed to decode cognitive states with strong biological inspiration. Unlike traditional artificial neural networks (such as the perceptron or multilayer perceptron), which drew only loose inspiration from neuroscience and abstracted away much of the brain’s complexity, BRICK integrates neuroscientific principles, such as fluctuating brain activity, neural oscillatory synchronization, and attentional memory to more closely emulate neurobiological processes.

Specifically, BRICK extends these neuroscience principles into the graph domain, employing a physics-informed model for brain rhythm identification. This model utilizes a Kuramoto model-inspired control mechanism to capture memory-guided neural synchronization. This approach enables the development of a neurobiologically inspired GNN, called BIG-NOS, which specifically addresses main limitations like over-smoothing while offering scalability across node and graph classification tasks.

Extensive evaluation on publicly available neuroimaging and graph datasets demonstrate that BRICK effectively synchronizes brain regions, yielding synchronization patterns closely aligned with specific cognitive states. BIG-NOS achieves state-of-the-art (SOTA) performance on a range of GNN benchmarks.

**Questions:**

- What are the pros of BRICK model w.r.t. to related work [9,10] that used Kuramoto model to describe the coupling between brain structure and function? What are the advantages of BRICK compared to the work in [44]?

- Can you clarify why equation 4 can be interpreted as gradient flow? I guess that in line 185 the authors want to provide intuition behind eq. 5 not 4.

- What kind on nerural nertwork architecture  did the author use to o parameterize the natural frequency Ω (lines 201-202)?

**Ethical Concerns:**

["NO or VERY MINOR ethics concerns only"]

**Final Justification:**

Dear authors, thank you for the rebuttal. After carefully reviewing all the comments and your response, I am confirming my original rating.

**Limitations:**

Yes

**Paper Formatting Concerns:**

No Paper Formatting Concerns.

**Quality:**

3

**Strengths And Weaknesses:**

Strengths
- Originality.
The main strength of this paper is to introduce a new design paradigm for developing GNNs through the lens of coupled neural oscillators. The authors highlight a link between the neuroscience and machine learning, where the last one is used learn the parameters the Kuramoto model, where a new term is included to incorporate the concept of the attending memory.

- Empirical validation.
The claim related to BRICK ant to BIG-NOS are well-supported by the experimental results, that show:
   - BRICK ability to synchronize brain region. Especially even the experiments in the absence of ground truth, show the BRICK’s capability of identifying clusters of functional subnetworks.
  - BIG-NOSE achieve SOTA performance on node and graph classification showing robustness to over-smoothing and scalability issues.


Weaknesses:

Clarity. The main weakness is related to the clarity of the paper. Specifically, the two following points limits the readability of the paper.
- Related work.
     - It is not clear what are the differences between the authors' modified Kumamoto model in BRICK and the ones proposed in [9,10].
    -  It is not clear what are the differences between the Kumamoto model in BRICK and the work [44]. In that regard, the authors contribution is to put the spotlight on bridging the reciprocal relationship between neuroscience and AI through the lens of governing equations in the dynamical system.

- Methods. The readability will improve to incorporate more information about the Big-NOS network architecture.

[9] Joana Cabral, Etienne Hugues, Olaf Sporns, and Gustavo Deco. Role of local network oscillaions in resting-state functional c	onnectivity. Neuroimage, 57(1):130–139, 2011.

[10] Katerina Capouskova, Morten L Kringelbach, and Gustavo Deco. Modes of cognition: Evidence
from metastable brain dynamics. NeuroImage, 260:119489, 2022.

[44] Takeru Miyato, Sindy Löwe, Andreas Geiger, and Max Welling. Artificial kuramoto oscillatory neurons. arXiv preprint arXiv:2410.13821, 2024.

---

> ### Author Rebuttal · Authors · 2025-07-30
>
> We sincerely thank the reviewer for recognizing the originality of our work and the comprehensiveness of our experimental evaluation. We also appreciate the thoughtful feedback regarding **related work** and **methodology details**. Below, we carefully address each concern and provide clarifications or revisions where needed.
>
> ### *1. It is not clear what are the differences between the authors' modified Kuramoto model in BRICK and the ones proposed in [9,10]. What are the advantages of BRICK compared to the work in [44]?*
>
> **[Differences between our model and [9], [10], [44]]**
> **BRICK and [9]:**
> While both *BRICK* and Cabral et al. [9] adopt the Kuramoto model to explore brain dynamics, their **roles and applications** diverge significantly. In their work, the Kuramoto model is used as a forward simulator to reproduce biologically observed functional connectivity (FC) patterns from empirical structural connectivity (SC), with the goal of explaining emergent brain phenomena. In contrast, our *BRICK* repurposes the Kuramoto model as a computational module within a machine learning framework. Rather than passively simulating dynamics, *BRICK* actively learns to perform representation learning, reasoning, and predictive tasks on graphs using dynamic synchronization mechanisms.
>
> **BRICK and [10]:**
> For **methodological paradigm**, while *Capouskova* et al. [10] follow a data-driven analytical paradigm, using autoencoders and clustering to uncover latent cognitive states from empirical fMRI data, they encode BOLD phase coherence data using modern machine learning tools **without Kuramoto model**. Their focus lies in neuroscientific pattern discovery and interpretation. In contrast, *BRICK* introduces a physics-informed learning framework that embeds the principles of neural synchronization, specifically via the Kuramoto model, directly into the learning process.
>
> The **outcomes** of these two approaches are also distinct: *Capouskova* et al. generate empirical insights about cognitive brain states, whereas *BRICK* yields new machine learning models that not only excel in brain-related tasks but also generalize to broader graph-based AI problems.
>
> **BRICK and [44]:**
> Compared with [44], our *BRICK* framework offers two key advantages.
> * **First, neuroscientific grounding**: starting from fMRI-derived brain rhythms, we augment the Kuramoto core with a task-driven feedback controller $y_i$ that mimics cognitive control, thereby preserving biological interpretability.
>
> * **Second, domain-specific impact**: on real HCP fMRI data *BRICK* achieves the best task decoding and unsupervised parcellation, while on graph benchmarks it delivers competitive accuracy and remains resistant to oversmoothing even at 128 layers.
>
> Taken together, these properties make *BRICK* a biologically inspired but practically efficient module that bridges brain dynamics and general-purpose AI.
>
> ### _2. Methods. The readability will improve to incorporate more information about the Big-NOS network architecture._
>
>  **[Details of BIG-NOS]**
> Conceptually, *BIG-NOS* can be viewed as an extension of *BRICK* to graph data. By interpreting graph nodes as analogs of brain regions and the adjacency matrix as a proxy for coupling strength among regions, we seamlessly adapt the physics-informed architecture of *BRICK* to arbitrary graphs. Both models share the same governing equations rooted in Kuramoto synchronization, but differ in their data domains and learning objectives.
>
> In *BRICK*, the inputs consist of BOLD signals over brain regions along with structural or functional connectivity; in *BIG-NOS*, the inputs are general node features and graph topology. Likewise, the outputs diverge in form: *BRICK* predicts cognitive or disease-related brain states (analogous to graph-level classification), while *BIG-NOS* predicts node/graph labels in standard graph learning settings.
>
> **We are committed to include the formal network detail of *BIG-NOS* into Section 3.2 as part of the overall architecture description.**
>
> ### _3. Can you clarify why equation 4 can be interpreted as gradient flow?_
>
>  **[Gradient flow interpretation of Eq.(4)]**
>
> Gradient flow refers to a class of dynamical systems that evolve along the steepest descent direction of an energy functional $E$. The evolution equation takes the form
> $
> \dot{\hat{x}}_i = -\frac{\partial E}{\partial \hat{x}_i}
> $
>
> which guarantees that the energy $\dot{E}$ decreases over time, i.e.,
> $
> \dot{E} = \frac{dE}{dt} \leq 0
> $
>
> leading the system toward a local minimum of $E$. The energy function defined in Eq.(5)
> $
> E = -\sum_{i,j} w_{ij} \hat{x}_i^\top \hat{x}_j - \sum_i y_i^\top \hat{x}_i
> $
> encodes two key objectives:
>
> * The first term promotes pairwise alignment (**synchronization**) among oscillators $\hat{x}\_i$ and $\hat{x}\_j$ with strong coupling $w\_{ij}$.
>
> * The second term aligns each oscillator with its task-specific control pattern $y_i$ (**controlled synchronization**).
>
> Taking the gradient of $E$ with respect to $\hat{x}_i$ yields
> $
> \frac{\partial E}{\partial \hat{x}\_i} = -\sum_j w\_{ij} \hat{x}\_j - y\_i
> $,
>
> so that the negative gradient becomes
> $
> -\frac{\partial E}{\partial \hat{x}\_i} = \sum_j w\_{ij} \hat{x}\_j + y_i
> $
>
> Substituting this into Eq.(4) we obtain:
> $
> \dot{\hat{x}}_i = \omega_i + \gamma \phi_i \left( -\frac{\partial E}{\partial \hat{x}_i} \right)
> $
>
> Thus, Eq.(4) is considered a gradient flow because its dynamics are equivalent to evolving along the negative gradient direction of an energy functional $E$. Here, the operator $\phi$ does not destroy the gradient flow structure, but instead imposes structural constraints on the descent direction (e.g., ensuring motion remains on the unit sphere).
>
> ### _4. What kind on nerural nertwork architecture did the author use to parameterize the natural frequency Ω (lines 201-202)?_
>
> **[Parameterization of natural frequency Ω]**
> The parametrization of natural frequency Ω does not rely on a deep neural network. Instead, Ω is parameterized using **a set of learnable vectors** whose norms determine the rotation speed (i.e., natural frequency) of oscillators. In the forward pass, these frequency values are used to rotate the 2D oscillator states (like turning a point in a circle), simulating how each node evolves over time due to its intrinsic dynamics.
>
> [9] Joana Cabral, Etienne Hugues, Olaf Sporns, and Gustavo Deco. Role of local network oscillations in resting-state functional connectivity. Neuroimage, 57(1):130–139, 2011.373
> [10] Katerina Capouskova, Morten L Kringelbach, and Gustavo Deco. Modes of cognition: Evidence from metastable brain dynamics. NeuroImage, 260:119489, 2022.375
> [44] Takeru Miyato, Sindy Löwe, Andreas Geiger, and Max Welling. Artificial kuramoto oscillatory neurons. arXiv preprint arXiv:2410.13821, 2024
>
> **Should there remain any ambiguities or if the reviewer has further questions, we would be more than happy to engage in continued discussion and resolve any remaining issues.**

---

> > ### Comment · Reviewer_SVVA · 2025-08-05
> >
> > Dear authors, thank you for your rebuttal, which addresses my concerns. I  confirm my rating to 5.

---

> > > ### Author Response · Authors · 2025-08-05
> > >
> > > Thank you very much for your kind follow-up. We truly appreciate your constructive feedback and your positive assessment of our work.

---

### Official Review · Reviewer_NLZ3 · 2025-07-01

**Clarity:** 2
**Significance:** 2
**Originality:** 3
**Rating:** 4
**Confidence:** 3

**Summary:**

Inspired by neural oscillation mechanism, the authors propose novel deep learning methods that utilize neural oscillation, BRICK and BIG-NOS. BRICK was applied for brain fMRI datasets. Specifically, BRICK achieved the best performance in brain analysis tasks, compared to simple GNN baselines. BIG-NOS was applied for graph benchmark datasets, and the authors report its superior node and graph classification accuracies compared to simple GNN baselines. The authors argue that the proposed approaches represent a new graph learning mechanism with SOTA performance.

**Questions:**

See weaknesses

**Ethical Concerns:**

["NO or VERY MINOR ethics concerns only"]

**Final Justification:**

I raised concerns about the paper's unfair experimental setting for graph learning benchmarks. During rebuttal, the authors reported results with improved experimental settings and revised their argument accordingly. Additionally, the authors reported SOTA performance of the proposed method in more brain application benchmarks. Considering all those factors, I consider a weak accept to be a proper final score.

**Limitations:**

Yes, but the authors only discussed marginal limitations.

**Quality:**

3

**Strengths And Weaknesses:**

The **key strengths** include:

- [S1 novelty]. The proposed approaches seem somewhat novel.
- [S2 visualization]. The visualizations are among the most beautiful ones I have seen in AI conferences.

However, I note two **critical limitations** to the present work.

- [W1 poor experimental setting]. **Many of the reported baseline performances in Table 3 are substantially lower than those reported in eariler studies**. For instance, GCNII performance on Cora is reported as 79.92, whereas the other works generally report up to 86 in the same train/val/test split. In fact, I have run many experiments with the baselines before, and the reported performances are definitely not optimal. I assume this is due to poor choice of hyperparameters. According to Appendix A.2, the authors did not tune some of the key hyperparameters for all baseline methods, and some other key hyperparameters, such as learning rate, are not reported. With such poor experimental setting, I find it hard to trust the experimental outcomes.
- [W2 lack of method justification]. I am not convinced of the justification to join neural oscillation with graph learning. Due to the issue in W1, empirical justification is hardly achieved. No formal theories were developed to justify them. The motivation illustrated in the introduction section (e.g., node in a graph as a couple oscillator) is only speculative and have not been analyzed or supporte by the prior works. Thus, I am not convinced why the proposed method is necessary or important.

Besides, I also find some **notable limitations**.

- [W3 missing details]. Some of important details are missing. For instance, formal description of the proposed method BIG-NOS seems to be missing. Is the Governing Equation in Table 1 the exact functional form of BIG-NOS? If so, please clarify. Also, the Figure 3 results should be buttressed with numeric outcomes. With only the plots, it is hard to objectively discern how they support the authors’ claims.
- [W4 missing prior works]. There have been some previous works that connect neural oscillation with graph neural networks [1,2]. However, the authors cited none of them. The authors should clarify how the proposed method compare to those earlier works.
    - [1] Graph-Coupled Oscillator Networks, ICML 2022
    - [2] From Coupled Oscillators to Graph Neural Networks: Reducing Over-smoothing via a Kuramoto Model-based Approach, ICML 2024
- [W5 unsupported claims]. The authors claim for SOTA performance. However, all the baselines are outdated, with the most recent one published in 2021. Even if the issue raised in W1 is somewhat justified, there is no evidence to demonstrate that the proposed method is not SOTA.

---

> ### Author Rebuttal · Authors · 2025-07-31
>
> We sincerely thank the reviewer for acknowledging the novelty and the value of our work. Below, we provide detailed responses to all concerns.
> ### *[W1 poor exp setting]，[W5 unsupported claims]*
> **[Exp setting and model selection]**
>
> We acknowledge that some of the reported accuracy for baseline models are lower than those in prior studies that specifically optimized each model/dataset. However, our intention was to ensure a **fair and controlled comparison** across all models under **uniform exp settings**, particularly important consideration in neuroscience, where the focus lies in uncovering mechanistic and functional insights.
>
> For all models, we fix major hyperparams as hidden dim 256;  #layers 4 (2 for GCN/GAT); #epochs 1500; lr 5e-4 – 1e-3; weight decay 5e-4. We will report all the details in final draft.
>
> While these settings may not be optimal for every model, e.g., GCNII perform best with 32–64 layers on Cora,which is not applicable for GCN or GIN due to over-smoothing or instability. Our choice thus ensures comparability under identical architectural constraints. We would also like to emphasize that the accuracy of GCN on Cora in our setting (81.66%) **is highly consistent** with prior reports (e.g., 81.5% in GRAND), which validates our implementation.
>
> In addition, since **our main focus is on neuroscience-related applications aligns with the Neuroscience and cognitive science track**, we prioritize baseline models that have been validated in computational neuroscience, such as GCN, GAT, GIN etc. [1]. Nevertheless, we have extended our comparisons by including GTN (2019), GRAND (2021), GraphCON (2022) and KuramotoGNN (KGNN) (2024). The results show that our models achieve significant improvements in brain datasets and retain strong performance in general graph benchmarks.
>
> **Brain states identification:**
> |Acc(%)|HCP-A|HCP-YA|HCP-WM|
> |-|--|-|-|
> |GRAND|86.18±1.79|47.36±0.67|33.27±2.63|
> |GTN|82.56±0.87|53.42±1.21|30.68±0.36|
> |GraphCON|87.87±2.29|66.29±1.04|44.12±2.1|
> |KGNN|85.53±1.68|45.68±1.12|35.79±1.54|
> |BRICK|95.55±0.77|84.20±1.60|89.22±1.71|
>
> **Node classification:**
> |Acc(%)|Texas|Wisconsin|Actor|Squirrel|Chameleon|Cornell|Citeseer|Pubmed|Cora|
> |-|-|-|-|-|-|-|-|-|-|
> |GRAND|67.42±7.81|73.09±4.83|33.34±1.16|34.45±1.84|38.43±3.33|56.55±9.15|72.1±1.07|79.12±0.3|83.73±0.58|
> |GTN|74.52±10.2|68.89±11.9|35.42±1.5|36.45±1.72|55.84±4.08|58.81±15.09|68.8±1.69|76.90 ± 0.77|78.60±1.72|
> |GraphCON|82.43±4.72|83.72±4.48|35.13±1.38|26.9±2.17|33.75±3.77|74.59±2.48|62.9±5.95|71.21±7.36|62.73±5.79|
> |KGNN|71.75±5.57|69.43±6.18|31.5±0.82|35.02±1.67|38.5±2.78|61.16±9.16|72.06±1.53|67.24±0.36|76.57±1.4|
> |BIG-NOS|81.35±3.51|82.16±3.56|35.8±0.6|68.06±1.65|74.5±1.13|73.24±4.75|70.63±0.36|77.8±0.23|81.86±0.25|
>
> **Graph classification:**
> |Acc(%)|ENZYMES|PROTEINS|
> |-|-|-|
> |GRAND|28±6.67| 71.97±4.26|
> |GTN|20.15±2.27| 75.48±4.27|
> |GraphCON|44.83±3.53| 69.82±6.12|
> |KGNN|29.26±2.84| 63.76±5.17|
> |BIG-NOS|60±4.28| 75.02±2.61|
>
> ### *[W2 lack of method justification]*
>
> We now clarify the **motivation**, **justification**, and **necessity** of our method.
> Our primary focus is ***brain rhythm identification*** from **neural oscillations**, which are fundamental in coordinating communication and cognitive functions [2]. Moreover, Kuramoto model has long been used to study coupled neural oscillators [3]. **Building on these foundation**, *BRICK* explicitly models oscillatory dynamics of brain function by Kuramoto dynamics, which not only achieves strong performance in brain state decoding (Tab.2), but also yields interpretable, biologically meaningful synchronization patterns (Fig.3). We believe this justifies both the **necessity and novelty** of our approach in the perspective of computational neuroscience.
>
> Encouraged by the promising results on brain data, we sought to genelize neural oscillation principles to **general graph domains** in AI/ML, which share the same notion of coupled dynamical systems as in brain networks. We hypothesize that **learning distinctive graph representations** is analogous to **coordinating functional roles (graph-level) across brain regions (node-level)**.
>
> We also provide the following methodology justifications. If there are **specific aspects of the theoretical formulation** that the reviewer would like further clarification on, we would be more than happy to provide detailed explanations as needed.
>
> **Proof of Lyapunov conditions.**
> Let $\hat{\mathbf{x}} = [\hat{x}\_1, ..., \hat{x}\_N] \in \mathbb{R}^{N \times d}$ and recall dynamical system Eq.(4) and energy $E$ Eq.(5). Because $W = [w\_{ij}]$ is symmetric (undirected),
>
> $
> \nabla_{\hat{\mathbf{x}}} E = - (W \hat{\mathbf{x}} + \mathbf{y}), \quad \mathbf{y} = [y_1 \ldots y_N]
> $
>
> Define $u_i := y_i + \sum_j w\_{ij} \hat{x}\_j$. Then Eq.(4) may be rewritten as:
>
> $
> \dot{\hat{x}}_i = \omega_i + \gamma \phi\_i(u_i) = \omega\_i - \gamma \phi\_i(-u\_i) = \omega\_i - \gamma \phi\_i([\nabla\_{\hat{\mathbf{x}}} E]\_i)
> $
>
> Then the time derivative of $E$ is defined as:
>
> $
> \dot{E} = \left\langle \nabla\_{\hat{\mathbf{x}}} E, \dot{\hat{\mathbf{x}}} \right\rangle = \sum\_i \underbrace{ \langle [\nabla\_{\hat{\mathbf{x}}} E]\_i, \omega_i \rangle}\_{\text{(i)}} - \gamma \underbrace{\langle [\nabla\_{\hat{\mathbf{x}}} E]\_i, \phi\_i([\nabla\_{\hat{\mathbf{x}}} E]\_i) \rangle }\_{\text{(ii)}}
> $
>
> The $\omega_i$ induces a pure phase rotation; it is orthogonal to the gradient direction, so
> $
>  \langle[\nabla_{\hat{\mathbf{x}}}E]_i, \omega_i\rangle=0
> $
>
> And $\phi\_i(\cdot)$ is
> **(i)** odd: \$\phi\_i(-z) = -\phi\_i(z)\$, and
> **(ii)** monotone increasing:
> $
>  \langle \phi_i(z_1)-\phi_i(z_2), z_1-z_2\rangle\geq 0
> $.
> With \$z = \[\nabla\_{\hat{\mathbf{x}}} E]\_i\$, we have \$\langle z, \phi\_i(z) \rangle \geq 0\$, this therefore yields (ii) $\leq 0\$. Combining (i) and (ii),
>
> $
> \dot{E}=0-\gamma\sum_i\langle z, \phi_i(z)\rangle\leq0, \quad z=[\nabla_{\hat{\mathbf{x}}} E]_i
> $
>
> If \$W \succeq 0\$ and both \$w\_{ij}\$ and \$y\_i\$ are finite, then Eq.(5) is at most quadratic with a finite lower bound:
>
> $
> E(\hat{\mathbf{x}})\geq-\lambda_{\max}(W) \|\hat{\mathbf{x}}\|^2-\|\mathbf{y}\| \|\hat{\mathbf{x}}\| \geq c_1 \|\hat{\mathbf{x}}\|^2-c_2
> $,
> for suitable constants $c_1 > 0$, $c_2 \geq 0$.
>
> Eq.(5) and the bound above provide the classical Lyapunov conditions:
> $
> E(\hat{\mathbf{x}}) \geq c_1 \|\hat{\mathbf{x}}\|^2 - c_2, \quad \dot{E}(\hat{\mathbf{x}}) \leq 0
> $
>
> Hence **\$E\$ is a Lyapunov function** for the Eq.(4): it is lower-bounded and monotonically non-increasing along every trajectory, guaranteeing stability.
>
> ### *[W3 missing details]*
>
> **[Details of _BRICK_ and _BIG-NOS_]**
>
> The governing equation presents in Tab.1 for BIG-NOS is shared with BRICK. This alignment is intentional and central to our design, as it enables a seamless adaptation of BRICK from human brain to general graph learning tasks. We'll give more details in final version.
>
> **[Interpretation of Fig.3]**
>
> We now elaborate how the discovered patterns in Fig. 3 support neurobiological plausibility:
> * **Alignment with functional subnetworks (Fig.3a):**
> *BRICK* captures task-specific synchronization across functional brain network in HCP-A (e.g., Visual network regions (red) cluster clearly during the VISMOTOR task in the deeper feature $X^{(L)}$, indicating strong within-network coordination). It suggest that *BRICK* respects functional boundaries.
> * **Inter-subject consistency and task-relevant encoding (Fig.3b):**
> Subjects performing the same task (e.g., “SOCIAL” in HCP-YA) form well-separated synchronization in deeper $X^{(L)}$. This indicates BRICK captures consistent latent dynamics across subjects, supporting that neural phase alignment underlines cognitive state representation.
> * **Unsupervised functional differentiation (Fig.3c):**
> *BRICK* can produce meaningful cortical parcellations without supervision. Compared to spectral clustering, *BRICK* yields phase-based partitions that are more spatially aligned with known functional networks [4].
>
> To provide stronger quantitative evidence, we have already included numeric results in Tab.2 and Appendix A.4. However, we will also include the following numeric figures to support Fig.3:
>
> - Within-network vs. between-network phase variance for Fig.3a to quantify the degree of synchronization.
> - Purity and NMI to measure how well the unsupervised parcels in Fig.3c align with the existing functional network.
>
> ### *[W4 missing prior works]*
> We have carefully reviewed both papers and will include them in the paper with a discussion of differences.
>
> 1. **Compare with GraphCON**: GraphCON is based on **second-order** ODEs representing *damped* nonlinear oscillators. Its formulation is **inspired by mechanical oscillators** to simulate oscillatory dynamics. *BRICK* is explicitly **informed by neural oscillation**, which is **first-order and phase-based**, aiming to model functional coordination in brain dynamics and translate those principles into learnable graph architectures.
>
> 2. **Compare with KuramotoGNN**: Their work demonstrates the use of synchronization dynamics for improving message passing in GNN, thus mitigating oversmoothing. However, *BRICK* incorporates **vector-valued oscillators** and **adaptive control mechanisms**. we ground our model in **neuroscientific applications** and demonstrate its performance and interpretability on **neural data**, beyond generic graph benchmarks.
>
> [1] Dan, T. et al. (2024). Exploring the enigma of neural dynamics through a scattering-transform mixer landscape for riemannian manifold.
> [2] Fries, P. (2005). A mechanism for cognitive dynamics: Neuronal communication through neuronal coherence.
> [3] Cabral, J.et al. (2011). Role of local network oscillations in resting-state functional connectivity.
> [4] Yeo, B.T. et al. (2011). The organization of the human cerebral cortex estimated by intrinsic functional connectivity.
>
> We are glad to provide further clarification or discuss in any format that would be helpful.

---

> ### Comment · Reviewer_NLZ3 · 2025-08-04
>
> Dear authors,
>
> Thank you for your rebuttal. I have carefully read your rebuttal and would like to raise few more points and questions.
>
> **[On W1 poor experimental setting]**. First, I am still not convinced about the fairness and rigor of experimental setting.
>
> - I agree that some baseline performance (e.g., GCN) seem reasonable.
> - *on fairness*: However, the authors used unreasonably detrimental hyperparameters for GCNII (specifically, small number of layers), and thus, I do not consider it a fair comparison. If the authors want to focus on the specific hyperpameter space, I strongly encourage the authors to remove the baselines that were not intended to be used with the hyperparameters. Likewise, I think the updated experiment with the added baselines is misleading as well, as both GraphCon and Grand have been reported to perform best at layers deeper than the ones the authors used.
> - *on the focus of the paper*: While the authors claimed that their focus is on “neuroscience-related applications aligns with the Neuroscience and cognitive science track”, they reported more results about graph learning (2 Tables and 1 Figure; 12 benchmark datasets) than those of neuroscience applications (1 Table and 1 Figure; 3 benchmark datasets). Moreover, in lines 314-315, the authors claim that their method “achieves SOTA performance on both heterophilic and homophilic graphs”. With such a heavy focus on graph learning, I must insist that the authors should not sidestep from the issue I raised (W1 poor experimental setting) by claiming they are not the focus. Besides, the reported performances of the proposed method is far away from SOTA performance. I encourage the authors to avoid using misleading claims.
>
> **[On W2 lack of method justification]**.
>
> - I understand that the authors designed their method by leveraging Kuramoto model and neural oscillation, which understandably may improve “brain learning”.
> - However, the authors’ motivation goes beyond brain learning. Specifically, they aimed to “rethink graph learning through the lens of coupled neural oscillators” (line 58) and design “a new graph learning mechanism via neural oscillatory synchronization” (line 90).
> - Why is leveraging neural oscillation and kuramoto model important for “graph learning”? As I stated earlier, the only motivation that the authors provided is speculative (”each node acts as a coupled oscillator, evolving through interactions governed by the graph topology”; line 78-79). In summary, the link between proposed method’s design principles and graph learning is unclear and unjustified.
> - Thus, contrary to the authors’ claim, I am not convinced this method is important for graph learning.
>
> In the future version of the paper, I encourage the authors to either (1) substantially reduce their focus on graph learning or (2) implement more rigorous and standard experimental settings accepted in graph learning domain, with a clear demonstration of the superiorities introduced by the proposed method over existing graph learning methods. In the current version, with such a heavy focus on graph learning and its poor justification/experiments, I cannot recommend acceptance.
>
> Sincerely,

---

> > ### Author Response · Authors · 2025-08-05
> >
> > **We sincerely thank the reviewer for the thoughtful and detailed comments regarding our experimental setup and the focus of the manuscript.**
> >
> > ### **[On W1 poor experimental setting]**
> >
> >
> > **[On experimental fairness]**
> >
> > We fully agree that the choice of hyperparameters plays a critical role in determining the performance of these models, and we appreciate the opportunity to clarify our decisions and revisions.
> >
> > To ensure faithful and fair comparisons, we closely followed the *official implementations* and *default configurations* provided by the original authors.
> >
> > * For all models whose original codebases include support for specific datasets along with recommended hyperparameters, we just re-run the code on these datasets and directly report their performance *without any modification (for Citeseer, Pubmed, Cora, we re-run the code 5 times to get mean±std)*. These include:
> >
> >   * **GCNII**: Texas, Wisconsin, Squirrel (not officially supported, but we followed the Chameleon setting as both are from *WikipediaNetwork*), Chameleon, Cornell, Citeseer, Pubmed, Cora
> >   * **GRAND**: Citeseer, Pubmed, Cora
> >   * **GraphCON**: Texas, Wisconsin, Cornell, Citeseer, Pubmed, Cora
> >   * **KGNN**: Texas, Wisconsin, Cornell, Citeseer, Pubmed, Cora
> >
> >   *(*We mark results using official hyperparameters in **bold** in the table.)*
> >
> > * For datasets that are **not covered by the official implementation**, we report results as described in our submission.
> >
> > |Model|Texas|Wisconsin|Actor|Squirrel|Chameleon|Cornell|Citeseer|Pubmed|Cora|
> > |-|--|-|-|-|-|-|-|-|-|
> > |GCN|58.65±3.64|52.75±6.35|28.38±0.96|28.87±1.5|39.36±1.93|45.14±4.84|70.62±0.84|77.76±0.50|81.66±0.50|
> > |GIN|57.03±5.98|47.84±5.20|25.91±1.07|25.16±2.17|32.17±1.75|48.65±10.61|58.28±3.09|71.40±2.10|70.76±2.66|
> > |GAT|56.49±5.85|53.53±7.60|29.05±0.80|30.08±1.03|43.16±1.56 |51.62±4.75|70.38±0.84| 76.80±0.75| 80.14±1.13|
> > |GCNII|**74.86±6.29**|**73.73±4.66**|33.80±1.41|**38.74±1.20**|**59.14±3.49**|**74.86±6.29**| **73.14±0.23**|**80.12±0.36**| **84.34±0.23**|
> > |GCNII*|*(**76.76±6.30**)*|*(**81.37±4.58**)*|*(**41.07±1.33**)*|*(**62.28±2.68**)*|*(**76.76±6.07**)*|||||
> > |GraphSAGE|78.92±5.51|79.61±7.55|34.88±1.19|36.90±1.03|48.03±2.22|70.81±3.15| 70.44±0.23|76.36±0.29|79.58±0.37|
> > |SAN |73.24±7.40|78.63±6.17|32.94±0.92|37.00±1.28|51.80±1.95|69.46±6.05|66.30±0.88|74.68±0.81|77.98±1.31|
> > |GRAND|67.42±7.81|73.09±4.83|33.34±1.16|34.45±1.84|38.43±3.33|56.55±9.15|**72.10±1.07**|**79.12±0.30**|**83.73±0.58**|
> > |GTN|74.52±10.20|68.89±11.90|35.42±1.50|36.45±1.72|55.84±4.08|58.81±15.09|68.80±1.69|76.90±0.77|78.60±1.72|
> > |GraphCON|**82.43±4.72**|**84.90±3.51**|35.13±1.38|26.90±2.17|33.75±3.77|**74.59±2.48**|**73.94±1.63**|**78.44±0.34**|**83.45±0.56**|
> > |KGNN|**71.75±5.57**|**69.43±6.18**|31.50±0.82|35.02±1.67|38.50±2.78|**61.16±9.16**|**72.06±1.53**|**67.24±0.36**|**76.57±1.40**|
> > |BIG-NOS|81.35±3.51|82.16±3.56|35.80±0.60|68.06±1.65|74.50±1.13|73.24±4.75|70.63±0.36|77.80±0.32|81.86±0.25|
> >
> > We have confirmed that our implementation for GraphCon on Cornell uses the optimal hyperparameter setting:`
> > 'cornell': {'model': 'GraphCON_GCN','lr': 0.00721,'nhid': 256,'alpha': 0,'gamma': 0,'nlayers': 1,'dropout': 0.15,'weight_decay': 0.0012708787092020595,'res_version': 1}`. However, we observed that the performance is lower than the result reported in the original paper. We remain open to discussion in case you have any insights or suggestions regarding this discrepancy.
> > A similar discrepancy is also observed for KGNN, despite using the optimal configuration in the code, and we also observed that reproducibility issues have been reported by others on the official GitHub.
> >
> > To offer a more balanced view, we have also computed average accuracy across **homophilic**, **heterophilic**, and **all datasets**, as shown in the updated results. BIG-NOS model delivers **strong** performance on **heterophilic datasets**, we find that it remains **competitive on homophilic datasets**, supporting its general effectiveness.
> >
> > |Model|GCN|GIN|GAT|GCNII|GCNII*|GraphSAGE|SAN|GRAND|GTN|GraphCON|KGNN|BIG-NOS|
> > |-|-|-|-|-|-|-|-|-|-|-|-|-|
> > |Hete. Avg|42.19|39.46|43.99|59.19|62.01|58.19|57.18|50.55|54.99 |56.28|51.23|69.18|
> > |Homo. Avg|76.68|66.81|75.77|79.20|79.20|75.46|72.99|78.32|74.77|78.61|71.96| 76.76|
> > |Total Avg|53.69|48.58|54.58|65.86|67.74|63.95|62.45|59.80|61.58|63.73|58.14| 71.70|
> >
> > All of our re-implemented models will be **released in our codebase** to ensure **reproducibility** and to foster **transparent discussion** around reproducibility and fairness in benchmarking.
> > We hope this clarifies our intentions and reinforces the fairness and integrity of our comparisons. We remain open to further suggestions and discussions from the community.

---

> > > ### Author Response · Authors · 2025-08-05
> > >
> > > **[On the focus and scope of the manuscript]**
> > >
> > > To ensure transparency and avoid any potentially misleading claims, we will revise the manuscript to remove generalized references to "state-of-the-art" (SOTA) performance. Instead, we now characterize our findings as *"promising,"* with a focus on demonstrating the feasibility and potential of our approach.
> > >
> > > That said, we would like to clarify that on heterophilic benchmarks such as **Actor**, **Squirrel**, and **Chameleon**, our method achieves the best performance among all compared approaches. In the revised manuscript, we will retain precise statements reflecting these leading results, while taking care to highlight the exploratory and proof-of-concept nature of our work.

---

> > > > ### Author Response · Authors · 2025-08-05
> > > >
> > > > ### **[On W2 lack of method justification]**
> > > >
> > > > **We sincerely thank the reviewer for raising this important and thought-provoking question regarding the justification of our design principles in the context of graph learning. We’d like to clarify the conceptual link between neural oscillations, the Kuramoto model, and graph learning is essential for strengthening the manuscript. Your comments have provided us with valuable guidance, and we are grateful for the opportunity to improve both the clarity and the broader impact of our work.**
> > > >
> > > > - **Motivation and Rationale:**
> > > >
> > > > The core motivation for leveraging neural oscillation, specifically, the Kuramoto model, in graph learning is to introduce a dynamic, interaction-based mechanism for information propagation and integration on graphs. In the brain, phase synchronization among coupled oscillators is thought to underpin robust, flexible communication between distributed regions. Following this notion, treating nodes as oscillators in graph learning allows us to model information exchange as a process of achieving local/global synchrony, naturally capturing both local structure and long-range dependencies in the graph.
> > > >
> > > > Conventional GNNs, based on message passing, can struggle with issues like over-smoothing and limited expressive power. By modeling node interactions as synchronization dynamics, we provide **an alternative mechanism** for feature propagation and aggregation, potentially leading to richer, more interpretable, and dynamically adaptive node/graph representations. Recent literature in network science and computational neuroscience suggests that synchronization-based mechanisms can capture critical aspects of network function, motivating their exploration within machine learning on graphs.
> > > >
> > > > - **Proof-of-Concept Scope:**
> > > >
> > > > We position this submission as a proof-of-concept study, intended to demonstrate the feasibility and promise of incorporating biologically inspired principles, specifically phase synchronization, into graph-based learning models. While we have a larger table to provide a more comprehensive empirical view for graph learning, this is intended to more thoroughly validate the core concept and generalizability across domains.
> > > >
> > > > - **Revision Action:**
> > > >
> > > > In light of your comments, we will revise the manuscript to clarify these points and to avoid any misleading statements. We now more explicitly emphasize the exploratory nature of our work.
> > > >
> > > > **If any part remains unclear or if the reviewer has further questions, we are more than happy to provide additional clarifications, explanations, or engage in further discussion in any form that may help.**

---

> > ### Author Response · Authors · 2025-08-06
> >
> > **Dear Reviewer,**
> >
> > We would like to sincerely thank you for the **time, effort, and expertise** you devoted to reviewing our submission. Your thoughtful feedback, particularly regarding the experimental settings, has helped us significantly improve the quality of our work. **We are truly grateful for the opportunity to improve our work through this revision process.**
> >
> > We hope that our latest revision has adequately addressed your concerns. If so, we would be most grateful if you would consider updating your score accordingly.
> >
> > Of course, if you have any remaining concerns or further suggestions, please do not hesitate to let us know, we would be more than happy to address them to the best of our ability.
> >
> > With sincere appreciation,
> > 23570 Authors

---

> > > ### Comment · Reviewer_NLZ3 · 2025-08-06
> > >
> > > Dear authors,
> > >
> > > I appreciate the clarification. I also appreciate the authors' attempts to address my concerns.
> > >
> > > I agree that the current empirical results on graph learning are promising. However, I generally do not consider 'promising results' to be good enough for a top-tier publication, especially with only thin experiments on brain data. Can the authors convince me why graph learning might need the proposed method? I would prefer a formal or empirical demonstration, but informal and intuitive explanations are welcome, too. The earlier explanations that the authors provided are unconvincing. For instance, the authors mentioned oversmoothing and limited expressivity of GNNs, but no evidence is provided that the proposed method may address those limitations. The authors also mentioned that the proposed method may capture 'both local structure and long-range dependencies', but there are countless GNNs that can achieve it.
> > >
> > > I am willing to raise my score, only if I end up agreeing that the proposed method may add value to the graph learning literature, despite its mediocre node classification performance.
> > >
> > > Best,

---

> ### Author Response · Authors · 2025-08-07
>
> We thank the reviewer for the constructive feedback and for considering a score adjustment based on the conceptual value of our work. Below, we provide a more focused response to your concerns.
>
> ### _1. However, I generally do not consider 'promising results' to be good enough for a top-tier publication, especially with only thin experiments on brain data._
>
> **we appreciate the opportunity to clarify the breadth and depth of our experimental validation on brain data.**
>
> In the current submission, we included three brain task datasets (HCP-A, HCP-YA, and HCP-WM) where we quantitatively demonstrate our model’s performance (Table 2), along with visualizations of synchronized neural dynamics in feature space (Figure 3a and 3b). We further explored the model’s ability to uncover biologically meaningful structures without supervision for functional parcellation (Figure 3c). Together, these results aim to show both **quantitative and qualitative** merits of our method in **brain applications**.
>
> **Beyond the main manuscript**, we have in fact conducted more extensive evaluations on neurological disease datasets, which were omitted due to space constraints, and we just focused on task-evoked fMRI datasets. We now summarize the results for three disease-related datasets (ADNI, PPMI, NIFD) here to provide a broader perspective:
>
> * **Alzheimer’s Disease Neuroimaging Initiative (ADNI)**: includes 135 resting-state fMRI samples from subjects diagnosed with Alzheimer’s disease (AD) or cognitively normal (CN) controls.
>
> * **Parkinson’s Progression Markers Initiative (PPMI)**: includes 173 samples spanning Parkinson’s disease (PD), SWEDD (scans without evidence of dopaminergic deficit), prodromal, and healthy controls.
>
> * **Neuroimaging Initiative for Frontotemporal Lobar Degeneration (NIFD)**: includes 1010 samples across a spectrum of frontotemporal dementia subtypes, including logopenic variant of primary progressive aphasia (LSD), behavioral variant (BV), progressive non-fluent aphasia (PNFA), semantic variant (SV), and cognitively normal (CON).
>
> | Model         | ADNI (Acc)       | PPMI (Acc)       | NIFD (Acc)       |
> | ------------- | ---------------- | ---------------- | ---------------- |
> | **GCN**       | 81.48 ± 7.77     | 57.14 ± 6.72     | 48.81 ± 1.55     |
> | **GIN**       | 79.26 ± 6.46     | 62.39 ± 5.71     | 49.90 ± 1.97     |
> | **GAT**       | 81.48 ± 7.87     | 58.96 ± 2.80     | 48.91 ± 2.06     |
> | **GCNII**     | 81.48 ± 7.77     | 59.46 ± 7.88     | 49.21 ± 1.70     |
> | **GraphSAGE** | 82.22 ± 6.37     | 61.83 ± 3.50     | 49.21 ± 1.49     |
> | **SAN**       | 82.96 ± 3.78     | 62.99 ± 7.16     | 49.91 ± 1.99     |
> | **GRAND**     | 81.48 ± 6.20     | 62.41 ± 7.14     | 43.46 ± 2.06     |
> | **GraphCON**  | 82.96 ± 5.54     | 60.12 ± 2.04     | 48.31 ± 1.35     |
> | **BRICK** | **83.05 ± 8.28** | **63.12 ± 4.59*** | **82.27 ± 6.42***  |
>
> On all three datasets, our model maintains strong classification performance. In particular, the results on PPMI and NIFD are statistically significant (*, $p < 0.01$, paired *t*-test), confirming the robustness of the observed improvements. This highlights the model’s potential in **handling complex, heterogeneous brain disorders**.
>
> We believe this is because _BRICK_ captures abnormal coordination patterns between brain regions, which is often a hallmark of neurodegeneration. The latent phase representation enables the model to detect subtle disruptions in functional connectivity that might be missed by conventional GNNs.
>
> **We hope this addresses the reviewer’s concern by demonstrating that *BRICK* is *consistently effective across a wide spectrum of brain datasets*.**

---

> > ### Author Response · Authors · 2025-08-07
> >
> > ### _2. Can the authors convince me why graph learning might need the proposed method? I would prefer a formal or empirical demonstration, but informal and intuitive explanations are welcome, too._
> >
> > Below, we provide both empirical evidence and intuitive explanations to support the motivation and usefulness of our model.
> >
> > • **[Strong performance on structurally heterophilic graphs and balanced performance across all the datasets]**
> >
> > As shown in updated table above, our model achieves the best performance on structurally heterophilic datasets **Squirrel** and **Chameleon**. In the table summarizing average performance below, our model demonstrates consistently good results across both homophilic and heterophilic graphs, **without major weaknesses**.
> >
> > |Model|GCN|GIN|GAT|GCNII|GCNII*|GraphSAGE|SAN|GRAND|GTN|GraphCON|KGNN|BIG-NOS|
> > |-|-|-|-|-|-|-|-|-|-|-|-|-|
> > |Hete. Avg|42.19|39.46|43.99|59.19|62.01|58.19|57.18|50.55|54.99 |56.28|51.23|69.18|
> > |Homo. Avg|76.68|66.81|75.77|79.20|79.20|75.46|72.99|78.32|74.77|78.61|71.96| 76.76|
> > |Total Avg|53.69|48.58|54.58|65.86|67.74|63.95|62.45|59.80|61.58|63.73|58.14| 71.70|
> >
> > **This balance is an encouraging signal:** while many existing models tend to specialize in one regime (e.g., GCN performs well on homophilic graphs), our model generalizes well to both homophilic and heterophilic graphs. We believe this robustness suggests that the underlying mechanism, neural oscillation, is more generalizable and structure-agnostic.

---

> > > ### Author Response · Authors · 2025-08-07
> > >
> > > • **[Resistance to over-smoothing: empirical and intuitive justifications]**
> > >
> > > In Figure 4 (we also provide the table below), we empirically demonstrate that _BIG-NOS_ remains stable up to 128 layers, with negligible degradation in performance, which is an indication of strong resistance to **oversmoothing**.
> > >
> > > |Layer|4|8|16|32|64|128|
> > > |-|-|-|-|-|-|-|
> > > |Acc(%)|81.0|80.4|81.4|81.9|81.9|81.9|
> > > |Pre(%)|81.95|81.59|82.45|82.80|82.82|82.82|
> > > |F1(%)|81.10|80.50|81.56|82.04|82.04|82.05|
> > >
> > > Intuitively, this robustness against **oversmoothing** stems from the oscillatory synchronization mechanism, which enables global coherence to emerge without collapsing all node features. This stands in contrast to traditional diffusive message passing, which tends to homogenize features as the network depth increases.
> > >
> > > Then, we provide a **heuristic analysis** illustrating how the coupling dynamics and feedback control in our model jointly mitigate excessive feature smoothing, by examining its simplified dynamics, steady-state solution and spectral response.
> > >
> > > **Simplified BRICK Dynamics**. To derive an interpretable steady-state solution and better understand the behavior of our model, we consider a **linearized simplification** of the _BRICK_ dynamics. We start with the original formulation:
> > >
> > > $
> > > \frac{d\hat{x}_i}{dt} = \omega_i + \gamma \phi\_i( y_i + \sum\_{j=1}^{N} w\_{ij} \hat{x}_j)
> > > $
> > >
> > > To simplify the analysis, we make two assumptions:
> > >
> > > 1. **Constant natural frequency** across all nodes: $\omega_i = \omega$, which can be absorbed into a baseline or ignored under steady-state assumptions.
> > > 2. **Linearization of the nonlinear projection**: We approximate the nonlinear function $\phi\_{\hat{x}_i}(\cdot)$ with an identity function, i.e., $\phi\_{\hat{x}_i}(\cdot) \approx \cdot$. This allows us to isolate the effect of network coupling and control term.
> > >
> > > The resulting dynamics become:
> > >
> > > $
> > > \frac{d\hat{x}_i}{dt} = \gamma( y_i + \sum\_{j=1}^{N} w\_{ij} \hat{x}_j)
> > > $
> > >
> > > For stability, we introduce a **dissipative force** to prevent unbounded growth, leading to:
> > >
> > > $
> > > \frac{d\hat{x}_i}{dt} = -\hat{x}_i + \gamma( y_i + \sum\_{j=1}^{N} w\_{ij} \hat{x}_j)
> > > $
> > >
> > > In matrix form, this can be compactly written as:
> > >
> > > $$
> > > \frac{d\hat{\mathbf{x}}}{dt} = -\hat{\mathbf{x}} + \gamma W \hat{\mathbf{x}} + \gamma \mathbf{y}
> > > $$
> > >
> > > This is a **linear consensus-like system**, where the $-\hat{\mathbf{x}}$ term acts as a stabilizing decay, and the $\gamma W \hat{\mathbf{x}} + \gamma \mathbf{y}$ term models network feedback and task-driven control.
> > >
> > > **Equilibrium Solution**. At steady state $\left( \frac{d\hat{\mathbf{x}}}{dt} = 0 \right)$, we obtain:
> > >
> > > $
> > > \frac{d\hat{\mathbf{x}}}{dt} = 0 \Rightarrow \hat{\mathbf{x}}^* = \gamma (I - \gamma W)^{-1} \mathbf{y}
> > > $
> > >
> > > The inverse exists provided the spectral radius of $\rho(\gamma W) < 1$ (i.e., $|\gamma \lambda_k| < 1$), where $\rho(\cdot)$ denotes the spectral radius, i.e., the largest absolute eigenvalue. This condition is typically satisfied in practice when $W$ is normalized.
> > >
> > > **Spectral Interpretation**. Let $W$ be symmetric and decomposed as $W = U \Lambda U^\top$, where $U$ is the orthonormal eigenvector matrix and $\Lambda = \text{diag}(\lambda_1, ..., \lambda_n)$. Projecting into the spectral domain (when $\gamma = 1$):
> > >
> > > $
> > > \hat{\mathbf{x}}^* = U (I - \Lambda)^{-1} U^\top \mathbf{y}
> > > $
> > >
> > > Let $\tilde{\mathbf{y}} = U^\top \mathbf{y}$ and $\hat{\tilde{\mathbf{x}}}^* = U^\top \hat{\mathbf{x}}^*$, then:
> > >
> > > $
> > > \hat{\tilde{x}}_k^* = \frac{1}{1 - \lambda_k} \tilde{y}_k
> > > $
> > >
> > > Thus, each spectral mode is scaled by a transfer function:
> > > **$h(\lambda_k) = \frac{1}{1 - \lambda_k}$**
> > >
> > > **Comparison with Diffusion-based GNNs**. In standard diffusion-type GNNs (e.g., GCN), applying $M$ layers is equivalent to using a transfer function $e^{-M \lambda_k}$ in spectral space. This leads to exponential suppression of high-frequency signals (large $\lambda_k$), causing oversmoothing. In contrast, BRICK uses a transfer function that decays much slower:
> > > $
> > > h(\lambda_k) = \frac{1}{1 - \lambda_k}
> > > $,
> > > which corresponds to $1/\lambda$-level suppression, allowing **high-frequency, discriminative signals to persist**.

---

> > > > ### Author Response · Authors · 2025-08-07
> > > >
> > > > • **[Conceptual novelty and broader potential impact]**
> > > >
> > > > _BRICK_ is a proof-of-concept model that integrates neural oscillator theory into brain network analysis and graph learning. We are motivated by two goals:
> > > >
> > > > * Explore new learning paradigms inspired by biological synchronization
> > > > * Enhance interpretability in both neuroscience and general graph domains
> > > >
> > > > On brain datasets, _BRICK_ already achieves strong predictive performance and insightful phase-based visualizations. On graph benchmarks, it delivers promising results with seamless adaptation. We believe this success can inspire future work on leveraging neural resonance and dynamic coordination mechanisms to design more interpretable and robust GNNs.
> > > >
> > > > In summary, our model contributes to brain network analysis by providing a biologically grounded framework for modeling neural dynamics, and to graph learning by introduce a new pwespective on addressing challenges such as heterophily and oversmoothing. We hope this convinces the reviewer of its relevance and potential.
> > > >
> > > > **We welcome any further questions and will do our best to respond with additional details or clarifications as needed.**

---

> > > > > ### Comment · Reviewer_NLZ3 · 2025-08-07
> > > > >
> > > > > Dear authors,
> > > > >
> > > > > I appreciate the thorough response. I have carefully read it, and many of my concerns have been adequately addressed. However, I am slightly concerned about the NIFD dataset result. The performance gap between the proposed method and baseline GNNs (~50%) seems unreasonably high to me. For comparison, I searched for papers that used GNNs on the same dataset, but I couldn't find any papers. For the camera-ready version, please make sure that the reported results are fair and reproducible.
> > > > >
> > > > > Considering all factors, I decide to raise my score.
> > > > >
> > > > > Best,

---

> > > > > > ### Author Response · Authors · 2025-08-07
> > > > > >
> > > > > > ### **We sincerely thank the reviewer for the constructive feedback and, *in particular*, for the willingness to **raise the score**. We deeply respect the reviewer’s rigorous reading and thoughtful concerns, and we truly appreciate the time and effort invested in evaluating our work.**
> > > > > >
> > > > > > Regarding the NIFD dataset performance: we understand that the improvement over standard GNN baselines may appear large. We have already noticed this gap and have carefully re-examined our experiments and confirmed that the reported results are accurate and reproducible. In fact, we observe similar behavior on another task-based dataset (HCP-WM), where *BRICK* also outperforms GNN baselines by a notable margin. Below, we provide a more detailed explanation for this phenomenon:
> > > > > >
> > > > > > * **Topology Dependency of GNNs**: In all brain experiments, we use BOLD signals as node features and structural connectivity (SC) as the graph topology. In the HCP-WM dataset, different tasks from the same subject share the same SC. Since GNNs heavily rely on the topology structure for message passing, they tend to weaken distinctions between different cognitive states of the same subject, leading to reduced performance [1].
> > > > > >
> > > > > >
> > > > > > * **Neural Oscillation vs. Static Propagation**: Real neural coordination is not solely governed by static, instantaneous topology, but rather by **temporal dynamics**, such as synchronization and resonance [2, 3]. Standard GNNs are not designed to capture such time-evolving processes, while *BRICK* explicitly models **oscillatory dynamics** with feedback control. This enables it to simulate **stable coordination** across time and regions, particularly critical for tasks involving sustained activity like working memory [4].
> > > > > >
> > > > > >
> > > > > > * **Functional Network-Level Coordination**: Working memory engages multiple regions in sustained co-activation (e.g., prefrontal, parietal, and temporal lobes). And in NIFD, **structural heterogeneity is strong** across a wide spectrum of frontotemporal dementia subtypes:
> > > > > >
> > > > > >      * *LPA*: affects the left temporoparietal junction
> > > > > >      * *bvFTD*: involves frontal cortex, insula, and amygdala
> > > > > >      * *PNFA*: involves the inferior frontal gyrus and anterior insula
> > > > > >      * *SV*: mainly atrophies in temporal poles
> > > > > >      * *CON*: healthy controls
> > > > > >
> > > > > >      These subtypes often involve **broad disruptions in the frontotemporal network**, which are not only spatially localized but also involve **long-range connections** [5, 6]. GNNs struggle to capture such patterns due to reliance on local aggregation. In contrast, ***BRICK* models global coupling through synchronization**, allowing it to dynamically detect these **distributed, non-local disruptions**, thus explaining its better performance on NIFD.
> > > > > >
> > > > > >
> > > > > >      Additionally, in smaller datasets like ADNI (135 samples) and PPMI (173 samples), all models are limited by sample size, making it harder to learn complex neural dynamics. However, in **NIFD (1010 samples)**, the larger dataset enables **BRICK** to fully leverage its dynamic synchronization to learn more **robust and generalizable patterns**. In contrast, GNNs suffer from **expressivity limitations** like we mentioned above and they fail to use the additional data effectively.
> > > > > >
> > > > > >
> > > > > > We hope this addresses the reviewer’s concern and reassures that the reported improvements are both fair and grounded in meaningful neurophysiological mechanisms. We are committed to ensuring reproducibility of the results and will incorporate all the results and discussions in the final version.
> > > > > >
> > > > > >
> > > > > > [1] Misra, Joyneel, et al. "Learning brain dynamics for decoding and predicting individual differences." *PLOS Computational Biology* 17.9 (2021): e1008943.
> > > > > > [2] Buzsáki, György. *Rhythms of the Brain*. Oxford University Press, 2006.
> > > > > > [3] Fries, Pascal. "Rhythms for cognition: communication through coherence." *Neuron* 88.1 (2015): 220–235.
> > > > > > [4] Funahashi, Shintaro. "Prefrontal cortex and working memory processes." *Neuroscience* 139.1 (2006): 251–261.
> > > > > > [5] Seeley, William W., et al. "Neurodegenerative diseases target large-scale human brain networks." *Neuron* 62.1 (2009): 42–52.
> > > > > > [6] Zhou, Juan, et al. "Divergent network connectivity changes in behavioural variant frontotemporal dementia and Alzheimer’s disease." *Brain* 133.5 (2010): 1352–1367.
> > > > > >
> > > > > >
> > > > > > ### **Once again, thank you for your thoughtful evaluation and support.**

---

### Note · Authors · 2025-08-11

### **We would like to sincerely thank AC and all the reviewers for your time, effort, and constructive feedback throughout the review process. The reviewers' thoughtful comments and suggestions have enabled us to significantly improve the overall quality of our manuscript.**

We are deeply encouraged by the reviewers’ enthusiasm for our novel paradigm, a biologically inspired mechanism that offers a fresh perspective to the AI field. **All reviewers found our approach to be novel and significant**:

- **Reviewer NLZ3**: “The proposed approaches seem somewhat novel. The visualizations are among the most beautiful ones I have seen in AI conferences.”
- **Reviewer SVVA**: “The main strength of this paper is to introduce a new design paradigm for developing GNNs through the lens of coupled neural oscillators.”
- **Reviewer zxED**: “This work presents a novel solution to address the oversmoothing issue of conventional GNNs. This system has novel designs in governing equation, learning mechanism, and attention mechanism compared to conventional GNN models. This model with good interpretability, and demonstrate interesting visualizations of synchronization patterns in Fig 3.”
- **Reviewer Rarc**: “This paper presents novel and significant work. It introduces a new perspective to the AI field, particularly in graph neural networks, by proposing a biologically inspired mechanism of brain rhythms for machine learning.

Regarding **Reviewer NLZ3**’s questions on the experimental setup and the motivation of our work, and **Reviewer Rarc’s** concerns on comprehensive comparison, biological plausibility, and the theoretical aspects, we have provided detailed and thorough responses, which they acknowledged and appreciated. **Both reviewers expressed that our rebuttal directly addressed their concerns and indicated their willingness to raise their scores**. **Reviewer SVVA confirmed a very positive score of 5, while Reviewer zxED did not engage in the discussion but had already given a positive initial score.**

Based on these meaningful discussions with the reviewers, **we commit to incorporating additional clarifications and updates in the final version of the paper regarding the experimental setup, comparisons, motivation, as well as the biological plausibility and theoretical aspects.**

### **Once again, we are grateful for AC's and the reviewers’ efforts in helping us strengthen our work.**

---

### Decision · Program_Chairs · 2025-09-17

**Decision:**

Accept (poster)

**Comment:**

The paper integrates a synchronization mechanism of neural oscillations with a graph representation learning framework. Motivated by addressing the oversmoothing issue of conventional GNNs, it proposed to utilize the brain rhythms in the artificial dynamical system. This method integrates a Kuramoto model with attending memory for modeling oscillatory synchronization in brain regions. The authors argue that the proposed approaches represent a new graph learning mechanism with SOTA performance.

Strengths identified by the reviewers include:

1. The approach seems novel. However as noted by a reviewer there is prior work integrating neural oscillation with graph neural networks which was not cited in the original manuscript.

2. Experiments on human brain datasets were appreciated by most reviewers. One reviewer however expressed some concerns which were addressed during rebuttal.

3. Good visualizations


Weaknesses identified include:

1. Important related prior work on neural oscillators not cited. The authors indicate in their rebuttal that they will cite missing work

2. A reviewer expressed concerns about the fairness and rigor of experimental setting, especially with respect to hyperparameters chosen.

3. More information needs to be reported on the computational cost of the proposed methods compared to existing methods

The authors did a good job addressing the reviewer concerns, which led reviewers to update their score. In particular the authors did a good job providing a better theoretical justification of their work, and a better justification of the biological plausibility, which is my understanding will be included in the final paper.

Overall the paper received 3 Accept ratings and 1 Borderline accept. Its score was significantly higher than the minimum threshold needed for acceptance so I recommend acceptance. However one of the reviewers did raise some valid concerns. The authors indicated in their rebuttal that they would address those concerns in their revised manuscript.